# Physics-informed Neural Operator for Pansharpening

**Xinyang Liu**[1,2][*] **Junming Hou**[1][*] **Chenxu Wu**[1]**, Xiaofeng Cong**[3]**, Zihao Chen**[1]
**Shangqi Deng**[5]**, Junling Li**[1]**, Liang-Jian Deng**[4][†]**, Bo Liu**[6][†]

[1] School of Information Science and Engineering, Southeast University
[2] School of Engineering Mathematics and Technology, University of Bristol
[3] School of Cyber Science and Engineering, Southeast University
[4] School of Mathematical Sciences, University of Electronic Science and Technology of China
[5] Institute of Artificial Intelligence and Robotics, Xi'an Jiaotong University
[6] Faculty of Computer Science and Control Engineering, Shenzhen University of Advanced Technology
`codex.lxy@gmail.com,junming_hou@seu.edu.cn`
`liangjian.deng@uestc.edu.cn,liubo@suat-sz.edu.cn`

## Abstract

Over the past decades, pansharpening has contributed greatly to numerous remote sensing applications, with methods evolving from theoretically grounded models to deep learning approaches and their hybrids. Though promising, existing methods rarely address pansharpening through the lens of underlying physical imaging processes. In this work, we revisit the spectral imaging mechanism and propose a novel physics-informed neural operator framework for pansharpening, termed PINO, which faithfully models the end-to-end electro-optical sensor process. Specifically, PINO operates as: (1) First, a spatial-spectral encoder is introduced to aggregate multi-granularity high-resolution panchromatic (PAN) and low-resolution multi-spectral (LRMS) features. (2) Subsequently, an iterative neural integral process utilizes these fused spatial-spectral characteristics to learn a continuous radiance field $L_i(x, y, \lambda)$ over spatial coordinates and wavelength, effectively emulating band-wise spectral integration. (3) Finally, the learned radiance field is modulated by the sensor's spectral responsivity $R_b(\lambda)$ to produce the desired fusion products. This physics-grounded fusion paradigm offers a principled solution for pansharpening in accordance with sensor imaging physics. Experiments on multiple benchmark datasets show that our method surpasses state-of-the-art fusion algorithms, achieving reduced spectral aberrations and finer spatial textures. Furthermore, extension to hyperspectral (HS) data demonstrates its generalizability and universality. The code is available at https://github.com/ez4lionky/PINO.

## 1 Introduction

Remote sensing satellites have become indispensable tools for Earth observation and monitoring, which span diverse civilian and military applications, such as precision agriculture, environmental monitoring, and mineral exploration [56, 67, 50, 49]. Unfortunately, existing optical satellite sensors struggle to directly acquire high-resolution multispectral (HRMS) observations. Instead, they typically acquire paired low-resolution (LR) MS and high-resolution panchromatic (PAN) images of the same scene. Pansharpening techniques aim to bridge the resolution gap by fusing complementary PAN and MS modalities to produce HRMS images, benefiting downstream remote sensing tasks.

---

[*]Co-first authors
[†]Corresponding Author

Over the past decades, the pansharpening community has witnessed a proliferation of methodologies, spanning from traditional model-driven algorithms to deep learning approaches and their hybrid variants. Despite notable advancements, most existing methods inadequately model the fundamental electro-optical process inherent to optical sensor systems, such as continuous radiance formation, optical filtering and sensor responsivity [59, 33, 79]. Traditional approaches, including component substitution and multiresolution analysis, typically operate on tensorized band data, applying heuristic transformations to align spatial details with spectral content. Deep learning approaches learn direct mappings from LRMS to HRMS. Both paradigms exhibit critical shortcomings: model-driven methods rely on simplified physical assumptions that often violate sensor-specific radiometric constraints, while data-driven approaches implicitly capture sensor characteristics through training data without explicit physical constraints. This disconnect from the underlying sensor imaging physics leads to critical challenges, including sensor-specific biases, spectral aberrations and spatial artifacts, restricting the applicability in real-world scenes.

In this work, we propose a physics-informed neural operator framework, named PINO, which unifies continuous radiance field modeling with learnable spectral responsivity, significantly boosting spatial-spectral fidelity and enhancing generalization across heterogeneous sensors and varying imaging conditions. The key procedures of PINO are as follows: First, we employ a spatial-spectral encoder to aggregate multi-granularity high-resolution PAN and LRMS features. Second, we exploit the fused spatial-spectral characteristics to learn a continuous radiance field over spatial coordinates and wavelengths, enabling the coexistence of sub-pixel spatial details and sub-band spectral nuances within the implicit neural representation. This process is implemented using an iterative kernel integral operator, effectively emulating band-wise spectral integration. Finally, we modulate the learned radiance field with the sensor's spectral responsivity, in accordance with the camera forward model, to generate high-resolution fusion products. We summarize the main contributions as follows:

- We propose a physically-grounded pansharpening framework that establishes a new paradigm bridging neural representations with physical sensor models, providing a principled solution applicable to multispectral and hyperspectral image fusion tasks.

- We employ an iterative kernel integral operator that leverages multi-granularity spatial–spectral features to learn a continuous radiance field over spatial coordinates and wavelength, enabling sub-pixel and sub-band fusion of different modalities while effectively emulating band-wise spectral integration.

- We introduce a learnable band-wise spectral responsivity modulation to simulate the sensor's spectral properties, allowing the simultaneous optimization of response functions.

- Experiments on multiple remote sensing benchmark datasets reveal that PINO consistently outperforms state-of-the-art methods in fusion capability, generaizalibity and adaptability.

## 2 Related Works

**Deep Learning-based MSI and HSI Fusion** In recent years, fueled by the availability of large-scale datasets, deep learning-based methods have achieved impressive progress in MS and HS image sharpening, significantly outperforming traditional model-driven approaches [16, 67, 24]. Typical techniques range from convolutional neural networks (CNNs) [78, 9, 25, 27, 19], global transformers [82, 23, 43, 73], to recently emerging generative diffusion models [55, 6]. In addition, growing interest has been directed toward hybrid approaches that integrate traditional fusion principles with deep learning architectures to enhance model interpretability [75, 84, 63, 35, 51]. While promising, these methods are largely grounded in mathematical models and rarely explore potential solutions from the physical imaging process. Concurrently, recent methods SSMNet and FBS-PS [42, 30] begin to incorporate sensor-related physics (e.g. band-separable properties) into learning-based fusion. However, they still fall short of modeling the full electro–optical imaging process.

**Implicit Neural Representation** Implicit neural representations (INRs) have drawn significant attention for their ability to encode continuous signals with high fidelity in a parametric form. This approach has been revolutionary in the domain of 3D computer vision, as exemplified by NeRF [57, 4], which demonstrated the potential of encoding intricate 3D structures with just 2D pose images. Recent advancements in INR architecture have led to the development of continuous image functions, which leverage the power of multi-layer perceptrons (MLPs) for tasks such as super-resolution (SR). For instance, LIIF [7] introduced a local implicit image function for SR, where the network performs

localized learning across the spatial domain to improve high-resolution reconstruction. Similarly, UltraSR [77] and DIINN [58] employ deep neural networks with residual connections and dual-interactive architectures, respectively, to further enhance decoding capabilities and spatial detail recovery. Furthermore, SSIF learns a continuous spatial–spectral representation for spectral image super-resolution [53], and SpectralNeRF performs physically based spectral rendering with a neural radiance field [36], illustrating how INR can extend beyond purely spatial signals. This line of work has naturally extended to the pansharpening and MHIF domain [54, 39, 13], where INRs have shown promise in enhancing the representation of spectral data. However, the current methods mainly use spatial coordinates or/and a simple band index to encode the continuous signal of spectral data.

**Image Super-resolution Neural Operator** Neural operators (NOs) have recently been introduced to learn mappings between infinite-dimensional function spaces [32, 3]. In contrast to classical neural networks, which typically operate on finite-dimensional Euclidean spaces or discrete sets, neural operators optimize network training within function spaces, offering superior nonlinear fitting and invariance to discretization [45, 38, 64]. As a resolution-invariant architecture, neural operators generalize effectively across different discretizations, rendering them highly suitable for tasks involving continuous or resolution-varying data. Wei et al. introduce SRNO, which integrates the function space mapping of neural operators with the efficient non-local modeling of Galerkin-style attention, offering a novel approach to continuous super-resolution [71]. DiffFNO designs a Weighted Fourier Neural Operator with a novel mode rebalancing mechanism to address the loss of high-frequency details in traditional counterparts caused by mode truncation [41]. HiNOTE introduces a hierarchical neural operator framework that effectively broadens the scope of neural operators to scientific data [46]. Very recently, He et al. apply neural operators to the hyperspectral pan-sharpening, developing a spectral-spatial continuous function mapping to solve the challenging scale generalization problem [20].

## 3 Methodology

In this section, we first introduce the preliminary of the spectral sensor imaging mechanism and then derive the proposed physics-informed neural operator framework. Subsequently, we elaborate on the implementation details of continuous radiance field learning, band-wise spectral responsivity modulation, optimization strategies and objectives within the proposed PINO.

### 3.1 Preliminary of Spectral Imaging Sensor Model

A multispectral or hyperspectral sensor records the scene's emitted or reflected radiance by focusing light through its optics and spectral filters onto a detector array. Along this optical path, the radiance spatial distribution undergoes radiometric, spatial and spectral transformations: lenses and dispersive elements shape the beam, filters select wavelength bands, detectors convert photons into an electronic signal, and an A/D converter quantizes that signal into digital numbers. Modeling these steps is essential to recover both high-resolution spatial detail and faithful spectra in subsequent fusion.

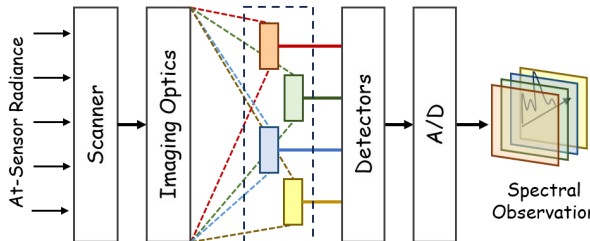

Figure 1: The key components and processes in an electro-optical spectral-imaging system.

As schematically depicted in Figure 1, this imaging pipeline operates as: (i) the scene radiance is collected and formed on the focal plane; (ii) it passes through band-selecting filters or prisms; (iii) detectors measure the filtered irradiance; (iv) the resulting analog signal is digitized. Let $L_i$ represent **spectral irradiance** on a detector located on the optical axis, it is related to the at-sensor radiance $L_i^s$ by the camera equation [60, 59]:

$$L_i(x, y, \lambda) = \frac{\pi \tau_o(\lambda)}{4N^2} L_i^s(x, y, \lambda) \quad (\mathrm{W} \cdot \mathrm{m}^{-2} \cdot \mu\mathrm{m}^{-1}), \tag{1}$$

where $N$ is the optics *f-number*, given by the ratio of the optical focal length divided by the aperture stop diameter. For simplicity, we assume unit geometric magnification between the scene and the

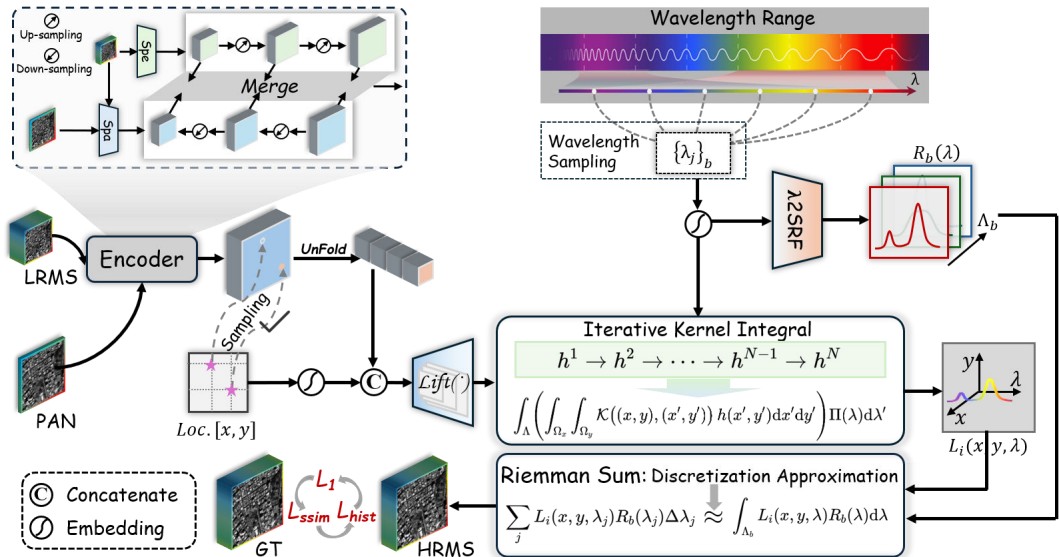

Figure 2: Overview of the proposed PINO. It features three key ingredients: (i) Spatial-spectral encoder: Aggregating multi-granularity spatial-spectral features; (ii) Iterative kernel integral operator: Learning a continuous radiance field over spatial coordinates and wavelength; (iii) Radiance Field Modulation: Modulating the learned continuous radiance field with the spectral responsivity to generate the desired HRMS images.

image plane, and use the same $(x, y)$ coordinate system for both the scene and image. The optical transmittance $\tau_o(\lambda)$ (excluding filters) remains high (>90%) and spectrally uniform in reflective systems, causing minimal spectral distortion.

We then introduced multispectral filters or wavelength dispersion elements, such as prisms, to split the incoming energy into distinct wavelength bands. Denote the combined filter transmittance and detector sensitivity by the **spectral responsivity** $R_b(\lambda)$, the signal intensity $I_{i,b}$ measured by the sensor in band $b$ is:

$$I_{i,b}(x, y) = \int_{\Lambda_b} L_i(x, y, \lambda) \, R_b(\lambda) \, d\lambda \quad (\text{W} \cdot \text{m}^{-2}), \tag{2}$$

where the integral is over the sensor's spectral range $\Lambda_b$. In practice, we discretize this integral via Riemann summation by sampling wavelengths $\{\lambda_j\}_b \in \Lambda_b$:

$$I_{i,b}(x, y) \approx \sum_j L_i(x, y, \lambda_j) R_b(\lambda_j) \Delta \lambda_j = \sum_j L_i(x, y, \lambda_j) R'_b(\lambda_j), \tag{3}$$

where $R'_b(\lambda_j)$ represents weighted spectral responsivity that incorporates both the original spectral responsivity and the wavelength sampling interval. Finally, the analogue signal $I_{i,b}(x, y)$ is converted to digital values via A/D conversion. In general, this model reveals a critical insight: *The sensor output is a weighted integral of continuous radiance*, motivating us to learn this mapping directly.

### 3.2 Physics-informed Neural Operator for Radiance Formation

To approximate the integral of continuous radiance, we first employ a transformer-based encoder to extract features from the input MS image $I^{MS} \in \mathbb{R}^{H \times W \times s}$ and PAN image $I^{PAN} \in \mathbb{R}^{H \times W}$ shown in Figure 2. Specifically, we use the bidirectional dilation transformer (BDT) architecture [12], which leverages dilated spatial self-attention and grouped spectral self-attention in a bidirectional hierarchy, expanding the spatial receptive field while capturing local spectral correlations. This allows the model to aggregate both global context and fine details across the multi-resolution inputs. The encoder thus outputs an intermediate feature map $\hat{F}$ that jointly encodes information from the upsampled MS image and the PAN image. More details about the encoder can be found in the Appendix A.3.3.

Given the fused feature map $\hat{F}$, we could query and unfold the feature to obtain each point-wise feature in the input resolution, which is a de facto standard for SR-related INRs [7, 39]. Meanwhile, we use periodic Fourier encoding [52] for both sampled spatial coordinates and wavelengths:

$$\Pi(v) = \rho\left(\left[\sin\left(\frac{v}{\sigma_0}\right), \cos\left(\frac{v}{\sigma_0}\right), \ldots, \sin\left(\frac{v}{\sigma_{T-1}}\right), \cos\left(\frac{v}{\sigma_{T-1}}\right)\right]\right),$$
$$\sigma_s = C_{\min} g^{\frac{t}{T-1}}, \quad t = 0, 1, \ldots, T-1, \tag{4}$$

where $v$ denotes the input, $T$ is the total number of grid scales and $g = \frac{C_{max}}{C_{min}}$. $C_{min}$, $C_{max}$ are predefined constants and present the minimum and maximum grid scale, $\rho$ could be parameterized by a MLP or a neural operator.

Consequently, we weight each neighboring feature by the fraction of its overlap with the query coordinates $(x, y)$ in the targeted-resolution to obtain an interpolated pixel-wise feature that captures local image content. As iullustrated in Figure 2, we concatenate these features with periodic fourier encodings of the same spatial coordinates, yielding the final fetched feature $F$. Based on $F$, we approximate the continuous radiance field $L(x, y, \lambda)$ by leveraging an iterative kernel integral operator with Galerkin-type integral form [5]:

$$h^0 = F, \quad h^1 \to h^2 \to \cdots \to h^{N-1} \to h^N, \tag{5}$$

with each hidden feature $h^n(x, y, \lambda), n = 1, 2, \cdots, N$, defined as:

$$g^{n+1}(x, y) = \int_{\Omega_x} \int_{\Omega_y} \mathcal{K}\big((x, y), (u, v)\big) \, h^n(u, v) \, \mathrm{d}u\mathrm{d}v \,, \tag{6}$$

$$h^{n+1}(x, y, \lambda) = \int_{\Lambda} g^{n+1}(x, y) \Pi(\lambda) \mathrm{d}\lambda', \tag{7}$$

$$\Rightarrow h^{n+1}(x, y, \lambda) = \int_{\Lambda} \left( \int_{\Omega_x} \int_{\Omega_y} \mathcal{K}\big((x, y), (u, v)\big) \, h^n(u, v) \, \mathrm{d}u\mathrm{d}v \right) \Pi(\lambda) \mathrm{d}\lambda', \tag{8}$$

after $N$ iterations, the hidden feature $h^N$ is projected to obtain the radiance field $L(x, y, \lambda)$. To make the computation tractable, we discretize the outer integral over $\lambda$ in Eq 8, approximating it via a finite summation, which we dubbed as wavelength modulation.

### 3.3 Band-wise Spectral Responsivity Modulation

While we have obtained the estimated radiance, as described in Section 3.1, we still need to integrate the radiance field through each band's responsivity to produce the recovered sensor measurements. Specifically, for each spectral band $b$, we parameterize its spectral responsivity function $R_b'(\lambda)$ as a learnable MLP layer that maps wavelengths to responsivity values. Note that this learnable responsivity function is shared across spatial coordinates and image contents, and the output of this MLP will be applied to perform band-wise modulation for estimated radiance, such that there are $s$ MLP layers in total. To ensure physical validity, we enforce non-negativity and smoothness constraints on the learned responsivity functions by adding a sigmoid activation function:

$$R_b'(\lambda_j, \theta_b) = \mathrm{Sigmoid}(\theta_b(\Pi(\lambda_j))), \tag{9}$$

where $\theta_b$ is learnable and parameterized by a MLP, $\lambda_j \in \{\lambda_j\}_b$. In summary, the learnable $R_b'(\lambda_j, \theta_b)$ lies within the range of $[0, 1]$ and could be regarded as a relative responsivity function [59].

Finally, we could predict the signal intensity and the pansharpened pixel value at the corresponding spatial coordinate and band:

$$\hat{I}_{i,b}(x, y) = \sum_j L_i(x, y, \lambda_j) R_b'(\lambda_j, \theta_b),$$
$$P_{i,b}(x, y) = \hat{I}_{i,b}(x, y) + P_{i,b}^{MS}(x, y) \tag{10}$$

where $P_{i,b}^{MS}(x, y)$ is the pixel value from the upsampled MS image at the corresponding spatial coordinates and band. The residual value could help the neural network to align the scale, as the physical dimension is not specified in the implicit neural representations.

## 3.4 Optimization Strategies and Objectives

Since learning implicit representations relies on sampling spatial coordinates, direct optimization can lead to the degradation or loss of spatial and semantic information from the input MS and PAN images. To mitigate this issue and build a robust foundation for spatial-spectral feature extraction, we adopt a two-stage training strategy. In the first stage, the encoder is trained independently using reconstruction losses, specifically L1 and SSIM, defined as:

$$\mathcal{L}_{s1} = \mathcal{L}_1 + \alpha_{s1}\mathcal{L}_{ssim}, \tag{11}$$

where $\alpha_{s1}$ is a trade-off hyperparameter that balances two loss terms, $L_1$ measures the mean of absolute error, $L_{ssim}$ measures structural similarity.

Then the encoder is fine-tuned in conjunction with PINO, which learns the continuous radiance field and spectral responsivity. Since the second-stage training involves uniform pixel sampling, local structural consistency is disrupted during the process. To address this, we incorporate the histogram loss $\mathcal{L}_{hist}$ [42] to align the statistical distributions of the predicted and reference pixel samples. Given the sampled HRMS pixels $\Psi_{\mathbf{H}}$ and corresponding reference pixels $\Psi_{\mathbf{G}}$, the loss for the second stage is defined as follows:

$$\mathcal{L}_{s2} = \mathcal{L}_1 + \alpha_{s2}\mathcal{L}_{hist},$$
$$\mathcal{L}_{hist} = \frac{1}{s} * \sum_{b=0}^{s-1} \left\| \mathbf{D}^i_{\Psi_{\mathbf{H}}} - \mathbf{D}^i_{\Psi_{\mathbf{G}}} \right\|_1, \tag{12}$$

where $\alpha_{s2}$ is introduced to balances the loss terms, $\mathbf{D}^i_{\Psi_{\mathbf{H}}}$ and $\mathbf{D}^i_{\Psi_{\mathbf{G}}}$ are the estimated 1D cumulative histogram vectors of the sampled pixels from the $i$-th fused spectral band and its corresponding reference sampled pixels, respectively.

## 4 Experiments

**Datasets, Metrics and Implementaion Details.** We assess the effectiveness of our method using data collected from the WorldView-3 (WV3), GaoFen-2 (GF2), and WorldView-2 (WV2) satellites, which are publicly available through the PanCollection dataset [10]. Owing to the absence of ground-truth references, the training datasets are simulated using observations from the original satellite imagery in accordance with Wald's protocol. In particular, WV2 samples are used for cross-sensor evaluation to further validate the model's generalizability and transferability. Reduced resolution evaluation is conducted using five established metrics: PSNR, SAM [80], ERGAS [68], Q2n and SCC [81]. Full resolution performance is assessed through three no-reference indicators: $D_\lambda$, and $D_s$, and HQNR [2]. All experiments are conducted on an NVIDIA GeForce GTX 4090 GPU, using the PyTorch framework. More implementation and training details can be found in Appendix A.3.4.

**Baselines.** We compare our model against several state-of-the-art DL-based methods to validate its superiority, including FusionNet [9], GPPNN [75], Fourmer [83], HFIN [62], and HOIF [85], PanMamba [21], LFormer [23], ADWM [28]. For a comprehensive evaluation, we also include three classical algorithms: BT-H [44], BDSD-PC [65], LRTCFPan [74].

### 4.1 Comparison With SOTA Methods

**Evaluation on Reduced Resolution Scene.** We first quantitatively assess the similarity between the fused images and the ground truth in reduced resolution scenarios. As shown in Table 1, our model surpasses state-of-the-art pan-sharpening methods across all evaluation metrics for both the WV3 and GF2 datasets. Figure 3 displays the RGB visualizations alongside their corresponding Mean Absolute Error (MAE) residues against the ground truth for both benchmarks. Our model's outputs exhibit minimal aberrations, as evidenced by its sparse MAE residue maps. These quantitative and qualitative findings together corroborate the superior fusion capability of our method.

**Evaluation on Full Resolution Scene.** We then evaluate our model on full resolution data to assess real-world applicability. As summarized in Table 1, it achieves the optimal outcomes on the majority of metrics for both the WV3 and GF2 datasets, mirroring the positive trends seen at reduced resolution. In particular, it delivers the highest HQNR values, surpassing both traditional and deep-learning baselines, which reflects superior spectral fidelity and spatial detail. These full resolution results confirm our method's robust generalization in practical settings.

Table 1: Quantitative results for reduced and full resolution WV3 and GF2 samples, comparing several representative state-of-the-art methods. Bold: Best; Underline: Second best.

| | Method | Reduced Resolution | | | | | Full Resolution | | |
|---|---|---|---|---|---|---|---|---|---|
| | | PSNR($\pm$ std) | SAM($\pm$ std) | ERGAS($\pm$ std) | $Q2^n$($\pm$ std) | SCC($\pm$ std) | $D_\lambda$($\pm$ std) | $D_s$($\pm$ std) | HQNR($\pm$ std) |
| WV3 Dataset | BDSD-PC [65] | 32.969$\pm$2.784 | 5.429$\pm$1.823 | 4.698$\pm$1.617 | 0.829$\pm$0.097 | 0.908$\pm$0.041 | 0.063$\pm$0.024 | 0.073$\pm$0.036 | 0.870$\pm$0.053 |
| | BT-H [44] | 33.080$\pm$2.880 | 4.920$\pm$1.425 | 4.579$\pm$1.496 | 0.832$\pm$0.094 | 0.925$\pm$0.024 | 0.057$\pm$0.023 | 0.081$\pm$0.037 | 0.867$\pm$0.054 |
| | LRTCFPan [74] | 33.613$\pm$2.839 | 4.737$\pm$1.412 | 4.315$\pm$1.442 | 0.846$\pm$0.091 | 0.927$\pm$0.023 | 0.018$\pm$0.007 | 0.053$\pm$0.026 | 0.931$\pm$0.031 |
| | FusionNet [9] | 38.042$\pm$2.592 | 3.325$\pm$0.698 | 2.467$\pm$0.645 | 0.904$\pm$0.090 | 0.981$\pm$0.007 | 0.024$\pm$0.009 | 0.036$\pm$0.014 | 0.941$\pm$0.020 |
| | GPPNN [75] | 38.571$\pm$2.776 | 3.055$\pm$0.610 | 2.306$\pm$0.587 | 0.914$\pm$0.087 | 0.984$\pm$0.006 | 0.028$\pm$0.010 | 0.038$\pm$0.016 | 0.935$\pm$0.022 |
| | Fourmer [83] | 38.268$\pm$2.727 | 3.236$\pm$0.681 | 2.419$\pm$0.665 | 0.911$\pm$0.090 | 0.984$\pm$0.005 | 0.022$\pm$0.010 | 0.035$\pm$0.004 | 0.944$\pm$0.013 |
| | HFIN [62] | 38.534$\pm$2.786 | 3.088$\pm$0.635 | 2.306$\pm$0.557 | 0.912$\pm$0.089 | 0.984$\pm$0.006 | 0.025$\pm$0.008 | 0.043$\pm$0.017 | 0.934$\pm$0.024 |
| | HOIF [85] | 38.352$\pm$2.855 | 3.186$\pm$0.643 | 2.385$\pm$0.668 | 0.913$\pm$0.086 | 0.982$\pm$0.007 | 0.039$\pm$0.021 | 0.039$\pm$0.010 | 0.924$\pm$0.026 |
| | PanMamba [21] | 39.012$\pm$2.818 | 2.914$\pm$0.592 | 2.184$\pm$0.521 | 0.920$\pm$0.085 | 0.986$\pm$0.005 | 0.018$\pm$0.007 | 0.031$\pm$0.011 | 0.952$\pm$0.015 |
| | LFormer [23] | 39.075$\pm$2.844 | 2.899$\pm$0.584 | 2.165$\pm$0.509 | 0.919$\pm$0.086 | 0.986$\pm$0.005 | 0.037$\pm$0.022 | 0.036$\pm$0.012 | 0.929$\pm$0.027 |
| | ADWM [28] | 39.170$\pm$2.878 | 2.914$\pm$0.589 | 2.145$\pm$0.531 | 0.919$\pm$0.086 | 0.986$\pm$0.005 | 0.024$\pm$0.010 | **0.029**$\pm$**0.015** | 0.948$\pm$0.021 |
| | PINO (ours) | **39.383**$\pm$**2.897** | **2.845**$\pm$**0.586** | **2.087**$\pm$**0.496** | **0.922**$\pm$**0.085** | **0.988**$\pm$**0.004** | **0.014**$\pm$**0.005** | 0.032$\pm$0.003 | **0.954**$\pm$**0.006** |
| GF2 Dataset | BDSD-PC [65] | 35.180$\pm$2.317 | 1.681$\pm$0.360 | 1.667$\pm$0.445 | 0.892$\pm$0.035 | 0.952$\pm$0.016 | 0.076$\pm$0.030 | 0.155$\pm$0.028 | 0.781$\pm$0.041 |
| | BT-H [44] | 36.054$\pm$2.236 | 1.649$\pm$0.360 | 1.528$\pm$0.409 | 0.918$\pm$0.025 | 0.957$\pm$0.015 | 0.060$\pm$0.025 | 0.131$\pm$0.019 | 0.817$\pm$0.031 |
| | LRTCFPan [74] | 37.599$\pm$2.331 | 1.298$\pm$0.312 | 1.272$\pm$0.343 | 0.935$\pm$0.030 | 0.964$\pm$0.012 | 0.033$\pm$0.027 | 0.090$\pm$0.014 | 0.881$\pm$0.023 |
| | FusionNet [9] | 39.639$\pm$2.270 | 0.974$\pm$0.212 | 0.988$\pm$0.222 | 0.964$\pm$0.009 | 0.981$\pm$0.005 | 0.0400$\pm$0.013 | 0.101$\pm$0.013 | 0.863$\pm$0.018 |
| | GPPNN [75] | 42.446$\pm$1.800 | 0.797$\pm$0.161 | 0.711$\pm$0.130 | 0.979$\pm$0.008 | 0.990$\pm$0.002 | 0.023$\pm$0.012 | 0.067$\pm$0.009 | 0.912$\pm$0.014 |
| | Fourmer [83] | 40.670$\pm$1.903 | 0.976$\pm$0.209 | 0.885$\pm$0.185 | 0.970$\pm$0.011 | 0.987$\pm$0.003 | 0.047$\pm$0.039 | 0.038$\pm$0.010 | 0.917$\pm$0.035 |
| | HFIN [62] | 42.189$\pm$1.752 | 0.843$\pm$0.148 | 0.735$\pm$0.126 | 0.977$\pm$0.011 | 0.990$\pm$0.002 | 0.027$\pm$0.020 | 0.062$\pm$0.009 | 0.912$\pm$0.018 |
| | HOIF [85] | 40.982$\pm$1.802 | 0.943$\pm$0.205 | 0.841$\pm$0.162 | 0.974$\pm$0.009 | 0.988$\pm$0.002 | 0.029$\pm$0.015 | 0.051$\pm$0.011 | 0.922$\pm$0.018 |
| | PanMamba [21] | 42.907$\pm$1.811 | 0.743$\pm$0.156 | 0.684$\pm$0.129 | 0.982$\pm$0.008 | 0.991$\pm$0.002 | 0.023$\pm$0.011 | 0.057$\pm$0.012 | 0.921$\pm$0.015 |
| | LFormer [23] | 44.196$\pm$1.800 | 0.648$\pm$0.130 | 0.578$\pm$0.112 | 0.985$\pm$0.007 | 0.993$\pm$0.002 | 0.021$\pm$0.010 | 0.050$\pm$0.008 | 0.930$\pm$0.013 |
| | ADWM [28] | 43.884$\pm$1.714 | 0.672$\pm$0.130 | 0.597$\pm$0.107 | 0.985$\pm$0.006 | 0.993$\pm$0.001 | 0.022$\pm$0.012 | 0.052$\pm$0.011 | 0.928$\pm$0.014 |
| | PINO (ours) | **44.705**$\pm$**1.819** | **0.615**$\pm$**0.129** | **0.544**$\pm$**0.107** | **0.987**$\pm$**0.006** | **0.994**$\pm$**0.001** | **0.016**$\pm$**0.009** | **0.018**$\pm$**0.008** | **0.967**$\pm$**0.009** |
| | Ideal value | $\infty$ | **0** | **0** | **1** | **1** | **0** | **0** | **1** |

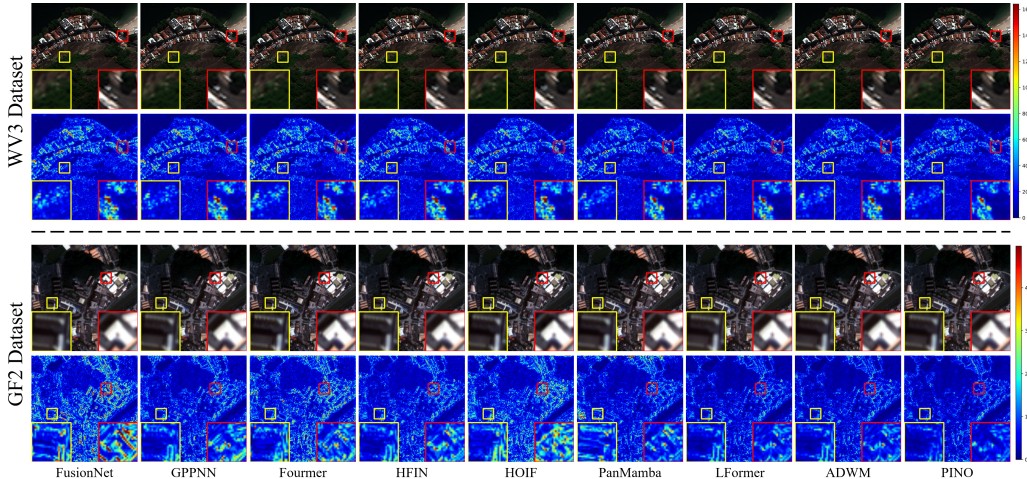

Figure 3: The visual results (top) and mean absolute error maps (bottom) of all compared DL-based methods on a reduced-resolution sample from WV3 and GF2 sensors, respectively.

**Generalizing to New Satellite Data.** We further evaluate adaptability by deploying the WV3-trained model on unseen WV2 samples without any fine-tuning. As reported in Table 2, our method outperforms all deep-learning baselines across all reduced and full resolution metrics, demonstrating its robust cross-satellite generalization and adaptability.

**Evaluation on Hyperspectral Datasets.** We further demonstrate our model's versatility by applying it to multispectral and hyperspectral image fusion (MHIF) task, which shares degradation principles with multispectral pansharpening. We benchmark against leading MHIF methods over the widely used CAVE dataset. As shown in Table 3, our approach outperforms all baselines across all metrics.

## 4.2 Ablation Studies

We first investigate the contribution of different core ingredients by performing ablation studies on the WV3 dataset with reduced resolution setting. To demonstrate the adaptability of our proposed

Table 2: Quantitative results for reduced and full resolution WV2 samples, comparing several state-of-the-art deep learning methods. Bold: Best; Underline: Second best.

| Method | Reduced Resolution | | | | | Full Resolution | | |
|---|---|---|---|---|---|---|---|---|
| | PSNR($\pm$ std) | SAM($\pm$ std) | ERGAS($\pm$ std) | $Q2^n$($\pm$ std) | SCC($\pm$ std) | $D_\lambda$($\pm$ std) | $D_s$($\pm$ std) | HQNR($\pm$ std) |
| FusionNet [9] | 28.734±2.460 | 6.426±0.860 | 5.136±0.515 | 0.796±0.074 | 0.875±0.013 | 0.052±0.029 | 0.056±0.015 | 0.894±0.019 |
| GPPNN [75] | 28.384±1.876 | 6.823±0.882 | 5.208±0.473 | 0.791±0.095 | 0.911±0.011 | 0.119±0.064 | 0.055±0.012 | 0.832±0.063 |
| Fourmer [83] | 28.924±2.418 | 6.224±0.459 | 4.915±0.348 | 0.821±0.078 | 0.898±0.017 | 0.038±0.027 | 0.085±0.017 | 0.879±0.017 |
| HFIN [62] | 30.093±2.208 | 5.467±0.681 | 4.411±0.447 | 0.835±0.083 | 0.918±0.008 | 0.059±0.045 | 0.039±0.013 | 0.905±0.054 |
| HOIF [85] | 29.917±2.242 | 5.490±0.655 | 4.517±0.457 | 0.834±0.085 | 0.908±0.010 | 0.068±0.046 | 0.043±0.058 | 0.894±0.088 |
| PanMamba [21] | 29.372±2.654 | 6.471±0.943 | 4.790±0.399 | 0.817±0.076 | 0.898±0.018 | 0.056±0.027 | 0.040±0.011 | 0.907±0.033 |
| LFormer [23] | 30.077±2.338 | 5.613±0.594 | 4.411±0.399 | 0.838±0.081 | 0.916±0.010 | 0.056±0.038 | 0.038±0.010 | 0.908±0.040 |
| ADWM [28] | 30.270±2.257 | 5.483±0.668 | 4.343±0.460 | 0.841±0.083 | 0.921±0.010 | 0.192±0.310 | 0.073±0.076 | 0.771±0.309 |
| PINO (ours) | 31.509±2.323 | 4.892±0.545 | 3.750±0.359 | 0.871±0.085 | 0.941±0.007 | 0.025±0.014 | 0.034±0.012 | 0.942±0.018 |
| Ideal value | $\infty$ | 0 | 0 | 1 | 1 | 0 | 0 | 1 |

Table 3: The average and standard deviation calculated for all the compared approaches on 11 CAVE examples simulating a scaling factor of 4 and 8. Bold: Best; Underline: Second best.

| Method | CAVE ×4 | | | | CAVE ×8 | | | |
|---|---|---|---|---|---|---|---|---|
| | PSNR($\pm$ std) | SAM($\pm$ std) | ERGAS($\pm$ std) | SSIM($\pm$ std) | PSNR($\pm$ std) | SAM($\pm$ std) | ERGAS($\pm$ std) | SSIM($\pm$ std) |
| Bicubic | 34.326±3.882 | 4.451±1.618 | 7.205±4.902 | 0.944±0.029 | 29.963±3.544 | 5.890±2.322 | 5.563±3.081 | 0.887±0.066 |
| CSTF-FUS [37] | 34.463±4.281 | 14.368±5.302 | 7.684±4.562 | 0.866±0.073 | 38.443±4.052 | 7.010±2.660 | 2.083±1.087 | 0.960±0.027 |
| LTTR [15] | 35.851±3.488 | 6.990±2.554 | 5.822±2.799 | 0.954±0.028 | 37.922±3.594 | 5.373 ±1.960 | 2.441 ±1.050 | 0.972±0.018 |
| LTMR [14] | 36.543±3.300 | 6.711±2.193 | 5.241±2.419 | 0.962±0.020 | 38.413±3.572 | 5.041±1.704 | 2.244 ±0.973 | 0.974±0.017 |
| IR-TenSR [76] | 35.608±3.446 | 12.295±4.683 | 5.715±2.899 | 0.944±0.026 | 36.787±3.638 | 12.865±4.979 | 2.667±1.399 | 0.943±0.030 |
| Fusformer [26] | 49.983±8.097 | 2.203±0.851 | 2.504±5.206 | 0.994±0.011 | 48.363±5.108 | 2.665±0.768 | 0.865±0.844 | 0.994±0.004 |
| DHIF [29] | 51.072±4.165 | 2.008±0.630 | 1.222±0.967 | 0.997±0.002 | 48.461±4.893 | 2.505±0.787 | 0.836±0.672 | 0.995±0.003 |
| PSRT [11] | 50.467±6.187 | 2.193±0.640 | 2.057±3.713 | 0.996±0.003 | 47.857±7.532 | 2.731±0.804 | 1.521±3.023 | 0.994±0.005 |
| 3DT-Net [47] | 51.376±4.179 | 2.161±0.695 | 1.137±0.996 | 0.996±0.003 | 48.985±7.015 | 2.299±0.653 | 1.185±2.227 | 0.996±0.003 |
| DSPNet [61] | 51.182±3.924 | 2.148±0.642 | 1.133±0.820 | 0.997±0.001 | 48.503±4.733 | 2.722±0.787 | 0.811±0.642 | 0.995±0.003 |
| QIS [86] | 52.218±4.213 | 1.977±0.599 | 1.023±0.806 | 0.997±0.001 | 49.441±4.988 | 2.545±0.772 | 0.733±0.625 | 0.996±0.003 |
| DCT [48] | 49.975±3.584 | 2.511±0.802 | 1.250±0.884 | 0.996±0.002 | 46.142±3.958 | 3.460±0.988 | 0.967±0.656 | 0.993±0.004 |
| MIMO [17] | 50.855±3.454 | 2.285±0.712 | 1.184±0.717 | 0.996±0.001 | 47.976±4.305 | 2.925±0.854 | 0.864±0.614 | 0.994±0.003 |
| PINO (ours) | 52.362±3.432 | 1.857±0.556 | 0.946±0.603 | 0.998±0.001 | 50.221±4.374 | 2.302±0.686 | 0.631±0.478 | 0.996±0.002 |
| Ideal value | $\infty$ | 0 | 0 | 1 | 1 | 0 | 0 | 1 |

PINO, we further apply it to different encoders and compare the performance gain. Moreover, we also reveal the importance of different optimization strategies and objectives. We also refer readers to Appendix A.6 for qualitative visualizations of the learned radiance maps and sensor responsivity, and to Appendix A.1, A.2 for theoretical analysis and physical-consistency details.

**Ablation Study of Core Design.** We investigate the integration of neural operators and radiance field estimation in the context of pan-sharpening. The radiance field plays a critical role in accurately representing the spectral information within produced MS images. To assess its contribution, we conduct ablation studies that explore different design configurations. Specifically, we consider the inclusion or exclusion of the neural operator as well as the impact of incorporating or omitting the radiance field estimation process. We also examine the role of spectral responsivity modulation, which refines the output to align with sensor-specific characteristics. Let "w/o no", "w/o sr" and "w/o wm" represent the results: "without neural operator", "without spectral responsivity" and "without wavelength modulation", respectively. As shown in Figure 4, we report three representative metrics "PNSR", "SAM" and "ERGAS" on the reduced resolution WV3 examples. We can find that the wavelength modulation is critical for the accurate radiance field estimation, without it, the pansharpening performance of PINO degrade largely. The neural operator and spectral responsivity are also essential for improving the performance. Overall, the results demonstrate that integrating these components improves fusion accuracy, resulting in better spectral integrity and spatial detail.

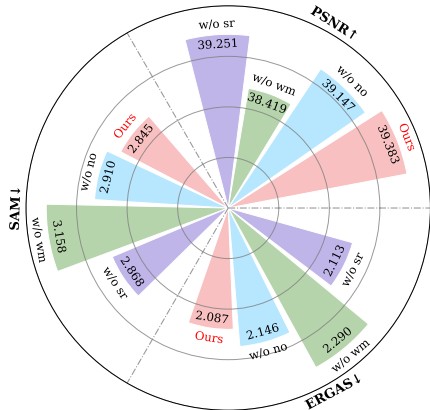

Figure 4: Ablation study of core design.

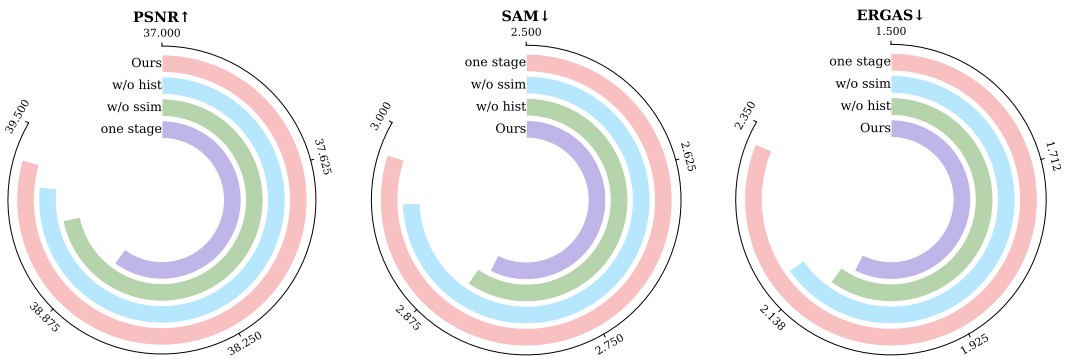

Figure 5: Ablation results for optimization strategies and objectives. "one stage", "w/o ssim" and "w/o hist" represent the results of "only training encoder with PINO in one stage", "without ssim loss" and "without histogram loss", respectively.

**Optimization Strategies and Objectives.** We investigate different optimization strategies, comparing single-stage versus two-stage training processes. In the single-stage training, the model is optimized as a whole, while in the two-stage approach, the encoder is first pretrained independently to extract spatial-spectral features. Then the encoder is fine-tuned in conjunction with PINO, which learns the continuous radiance field. We also explore the use of different loss functions, including SSIM loss and histogram loss, to ensure that both spatial and spectral integrity are preserved during the fusion process. Figure 5 compares the impact of different optimization strategies on fusion quality, where our method yields the optimal results, highlighting the efficacy of the selected techniques.

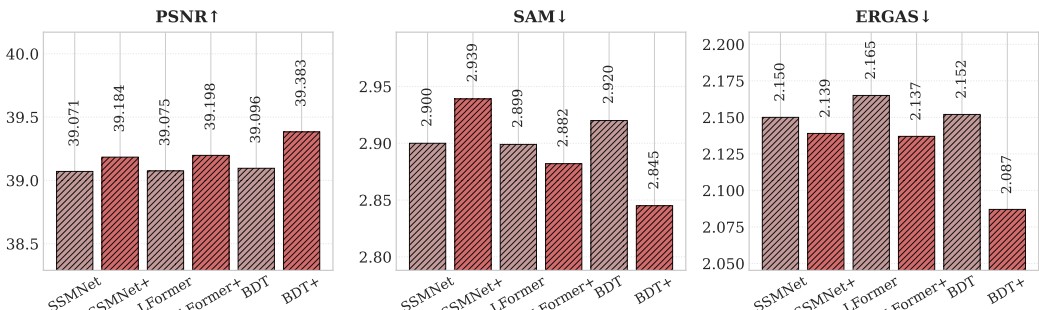

Figure 6: Ablation results for different encoder architectures with and without PINO. $X$ means original encoder architecture while $X+$ denotes corresponding encoder with PINO.

**Encoder Architecture.** We explore the influence of different encoder architectures on the performance of PINO. The encoder is responsible for aggregating multi-granularity features from the input high-resolution PAN and LRMS images. We conduct a comparative analysis of various encoder architectures, such as SSMNet, LFormer and BDT, focusing on evaluating the performance improvement when the encoder is combined with PINO compared to the encoder alone. The results in Figure 6 highlight the advantages of PINO-enhanced encoder architectures, which offer improved feature fusion and support more accurate pansharpening.

## 5  Limitations and Conclusion

Unlike conventional INRs developed for super-resolution tasks, in this paper, we mainly focus on the spectral imaging process and incorporate its physical sensor characteristics. We do not explore the use of arbitrary resolution settings for training and inference, which remains uncommon in current practices of the pansharpening community. Nevertheless, investigating this capability could greatly reduce the computation and parameters by unifying the redundant models and training for each targeted resolution. Moreover, cross-sensor or cross-dataset training is not explicitly addressed in our current design. The computational complexity and model size of our framework could also

be further optimized to enhance efficiency and scalability. In the wavelength modulation, we use uniform spectral sampling, which is simple and empirically effective but potentially suboptimal under strong wavelength variability; sensor-aware/adaptive schemes (e.g., denser samples where spectral responsivity or radiance changes rapidly) could better capture fine spectral detail.

We present a physically-grounded pansharpening framework that establishes a new paradigm bridging neural representations with physical sensor models, providing a principled solution applicable to multispectral and hyperspectral fusion tasks. At its core, our method first learns a continuous radiance field $L_i(x, y, \lambda)$ over spatial coordinates and wavelength using multi-granularity spatial-spectral features, effectively emulating band-wise spectral integration. By modulating this radiance field with the sensor's spectral responsivity $R_b(\lambda)$, it can generate high-fidelity fusion outputs. Extensive experiments on multiple benchmark datasets demonstrate that our approach outperforms state-of-the-art fusion algorithms in both quantitative metrics and qualitative comparisons.

## 6   Acknowledgements

This work is funded by National Natural Science Foundation of China (Grant No. 62203309 and 12271083), and Guangdong Basic and Applied Basic Research Foundation (Grant No. 2024A1515011333), and the Project of the Department of Science and Technology of Sichuan Province (Grant No. 2025YFNH0001).

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

# A    Appendix / Supplementary Material

This supplementary material provides additional insights into the background, methodologies, and experimental details outlined in our paper. It includes details about theoretical analysis and physical consistency of proposed physical modeling, datasets, metrics, the employed bidirectional transformer encoder, implementation and training. Furthermore, extensive qualitative visualizations are presented across standard pansharpening benchmarks and multispectral and hyperspectral image fusion, along with analyses of the learned radiance field and spectral responsivity, to substantiate the interpretability and physical grounding of our model. The provided information aims to enhance the reader's understanding of the intricacies involved in our research and its practical applications.

## A.1    Theoretical Analysis of Physical Modeling

We begin by revisiting and summarizing the electro–optical image formation model used throughout the paper and detailing how each component is instantiated within PINO, providing the context for the subsequent discussion on boundary behavior and physical consistency.

**1) Sensor physics modeling.** (i) Irradiance formation follows the camera equation (Eq. (1) in the main paper),

$$L_i(x, y, \lambda) = \frac{\pi \, \tau_o(\lambda)}{4N^2} \, L_i^s(x, y, \lambda), \tag{13}$$

which is realized within our spatial–spectral encoder. (ii) Band-wise spectral integration (Eq. (2) in the main paper) is

$$I_{i,b}(x, y) = \int_{\Lambda_b} L_i(x, y, \lambda) \, R_b(\lambda) \, d\lambda, \tag{14}$$

and in our PINO this integral is approximated via an *iterative* kernel integral operator (Eq. (8) in the main paper), while $R_b(\lambda)$ is a learnable, modulated function (Eq. (9) in the main paper). (iii) Discretization (Eq. (3) in the main paper) reflects sensor quantization:

$$I_{i,b}(x, y) \approx \sum_j L_i(x, y, \lambda_j) \, R_b(\lambda_j) \, \Delta\lambda_j = \sum_j L_i(x, y, \lambda_j) \, R_b'(\lambda_j). \tag{15}$$

The above formulas explicitly encodes key electro-optical process, and the proposed learning framework is built on this theoretical base.

**2) Neural operator as physics-informed solver.** The iterative Galerkin-style kernel integral operator acts as a functional approximator for the spectral integral in Eq. (2):

$$h^{n+1}(x, y, \lambda) = \int_{\Omega_x} \int_{\Omega_y} \mathcal{K}\big((x, y), (u, v)\big) \, h^n(u, v) \, du \, dv \, \cdot \, \Pi(\lambda), \tag{16}$$

where $\mathcal{K}$ captures spatial–radiance coupling and $\Pi(\lambda)$ provides implicit spectral basis functions (e.g., Fourier encoding); the wavelength discretization above mirrors sensor quantization. This iterative operator lets the model learn a continuous radiance field aligned with radiative-imaging physics.

**3) Spectral responsivity as differentiable physics.** Parameterizing each band's responsivity as:

$$R_b'(\lambda_j; \boldsymbol{\theta}_b) = \sigma\big(\theta_b(\Pi(\lambda_j))\big), \qquad R_b'(\lambda_j; \boldsymbol{\theta}_b) \in [0, 1], \tag{17}$$

which guarantees the non-negativity that aligns with the bounded nature of real sensor spectral responses; the differentiable sigmoid (with gradient $\sigma'(x) = \sigma(x)(1 - \sigma(x))$) avoids non-physical discontinuities, $\Pi(\lambda)$ is a spectral encoding (e.g., Fourier features), and the learnable mapping $\theta_b(\cdot)$ (e.g., MLP) adapts to sensor-specific bands.

## A.2    Boundary Conditions of the Radiance Field and Physical Consistency

Building on the above modeling, we next state the boundary assumptions for the continuous radiance field and clarify how physical consistency is enforced in the learned sensor model.

We assume *homogeneous Neumann boundary conditions* for the radiance field in Eq. (1) of the main paper, i.e.,

$$\left. \frac{\partial L_i^s(x, y, \lambda)}{\partial \mathbf{n}} \right|_{\partial\Omega} = 0, \tag{18}$$

which reflects the smooth transition commonly observed at image boundaries in natural and remote-sensing scenes. Such boundary smoothness is a standard modeling choice in image processing and remote sensing [8, 34, 72].

Although we do not explicitly encode boundary conditions into the network or loss, they can be *implicitly* learned from data due to the inductive bias of convolutional operations and padding. Prior work reports that networks trained for restoration or remote-sensing tasks learn smooth boundary transitions even without explicit constraints [87, 40]. Visualizations in Appendix A.6 further show that our learned radiance fields exhibit boundary smoothness closely matching the ground truth.

While the boundary behavior of $L_i^s$ is handled implicitly, we place an *explicit* physics prior on the sensor's spectral responsivity to ensure global physical consistency. Because responsivity is *sensor-specific* and independent of scene content, we enforce **non-negativity** and **smoothness** by parameterizing each band with a sigmoid activation. This guarantees learned spectral responses are bounded, smooth, and sensor-consistent, leading to radiance maps that preserve fine spatial structures and realistic wavelength-dependent variations; see Appendix A.6. Together with wavelength modulation and the neural operator, this yields a PINO that is both *physically consistent* and *spectrally faithful*.

## A.3 Datasets, Metrics and Implementation Details

### A.3.1 Datasets

We evaluate the pan-sharpening performance of our framework over several public benchmark datasets, including WorldView-3 (WV3), GaoFen-2 (GF2) and WorldView-2 (WV2). MS images within the WV3 and WV2 datasets contain eight spectral bands: coastal 397–454/400–450 nm, blue 445–517/450–510 nm, green 507–586/510–580 nm, yellow 580–629/585–625 nm, red 626–696/630–690 nm, red-edge 698–749/705–745 nm, near-IR1 765–899/770–895 nm, and near-IR2 857–1039/860–1040 nm. The corresponding PAN images observed from the same scene are single-channel images. The GF2 dataset pairs a single-band PAN image with four MS channels covering blue 450–520 nm, green 520–590 nm, red 620–690 nm and NIR 770–890 nm. Notably, due to the absence of ground-truth (GT) images, we generate reduced-resolution MS–PAN training and testing pairs following Wald's protocol [69].

To evaluate the versatility of our model, we further conducted experiments on the multispectral and hyperspectral image fusion (MHIF) task by using the CAVE dataset. The CAVE dataset comprises 32 Hyperspectral Images (HSIs) with 31 spectral bands spanning from 400 nm to 700 nm at 10 nm intervals. We randomly selected 20 images for training and used the remaining 11 for testing (same as [39, 13]). The RGB visualization of the test set is shown in Figure 7.

In our experiments, we uniformly sample and normalize the wavelength with min-max scaling according to the specific spectral range of each band:

$$\tilde{\lambda} = 2 \frac{\lambda - \lambda_{\min}}{\lambda_{\max} - \lambda_{\min}} - 1, \tag{19}$$

where $\lambda_{min}$ and $\lambda_{max}$ represent the minimum and maximum value of spectral wavelength for the corresponding band. For example, $\lambda_{min} = 397$, $\lambda_{max} = 454$, $\lambda \sim \mathcal{U}(\lambda_{min}, \lambda_{max})$ for the WV3 coastal band spans 397–454 nm, and the sampled wavelength $\tilde{\lambda}$ is normalized to $[-1, 1]$.

### A.3.2 Metrics

Following standard evaluation protocols in the pansharpening community, we assess the quality of fused images using both reference-based and no-reference metrics. For reduced-resolution quality assessment, we adopt the Peak Signal-to-Noise Ratio (PSNR) [22], Spectral Angle Mapper (SAM) [80], Relative Dimensionless Global Error in Synthesis (ERGAS) [68], Spatial Correlation Coefficient (SCC) [81], Structural Similarity (SSIM) [70] and the generalized $Q2^n$ index [18], where $n$ corresponds to the number of spectral bands (e.g., Q8 for 8-band data and Q4 for 4-band data). For full-resolution no-reference evaluation, we utilize three widely accepted indicators: the Hybrid Quality with No Reference (HQNR) [1], Spectral Distortion Index ($D_\lambda$), and Spatial Distortion Index ($D_s$) [66]. The formal definitions of these metrics are outlined below.

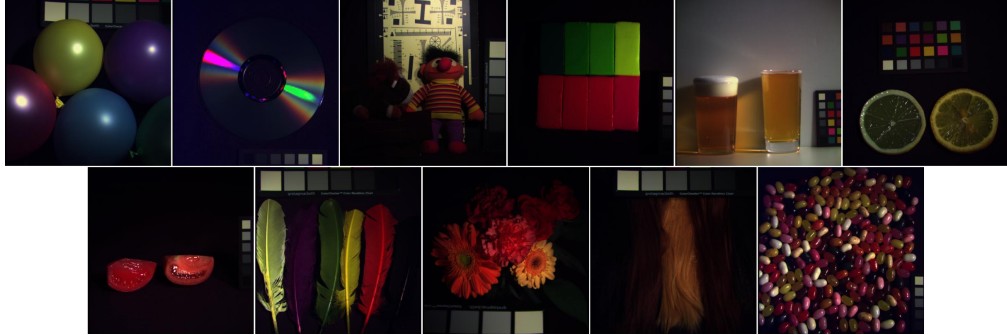

Figure 7: The RGB visualization of CAVE testing samples.

(1) Peak Signal-to-Noise Ratio (PSNR): PSNR evaluates the pixel-wise fidelity of each reconstructed band by comparing it to the ground truth. We compute the average PSNR over all $B$ bands as

$$\text{PSNR} = \frac{1}{B} \sum_{b=1}^{B} \text{PSNR}(I_b, \hat{I}_b), \tag{20}$$

where $I_b, \hat{I}_b \in \mathbb{R}^{H \times W}$ are the $b$-th bands of the reference and fused images, and

$$\text{PSNR}(I_b, \hat{I}_b) = 20 \log_{10}\left(\frac{\max(I_b)}{\sqrt{\text{MSE}(I_b, \hat{I}_b)}}\right), \tag{21}$$

where MSE represents Mean Square Error, and $\max$ represents the maximum value.

(2) Spectral Angle Mapper (SAM): SAM evaluates the angular difference between spectral vectors of the fused image and ground truth (GT), with an ideal value of 0. It is defined as:

$$\text{SAM} = \frac{1}{B} \sum_{b=1}^{B} \arccos\left(\frac{\mathbf{r}_b \cdot \hat{\mathbf{r}}_b}{|\mathbf{r}_b|_2 \cdot |\hat{\mathbf{r}}_b|_2}\right), \tag{22}$$

where $B$ is the number of spectral bands, $\mathbf{r}_b$ and $\hat{\mathbf{r}}_b$ denote the $b$-th spectral vector from the GT and fused image, respectively.

(3) Relative Dimensionless Global Error in Synthesis (ERGAS): ERGAS measures the global radiometric distortion, ideally approaching zero. It is computed as:

$$\text{ERGAS} = 100 \cdot s \cdot \sqrt{\frac{1}{B} \sum_{b=1}^{B} \frac{\text{RMSE}(\mathbf{r}_b, \hat{\mathbf{r}}_b)}{\mu_b}}, \tag{23}$$

where $s$ is the spatial resolution ratio between PAN and MS images, $\mu_b$ is the mean of the $b$-th GT band, and RMSE denotes the root mean square error.

(4) Spatial Correlation Coefficient (SCC): SCC quantifies the similarity of spatial details between the fused image and GT using a high-pass filter and correlation coefficient (CC). The specific calculation of SCC includes two steps: 1) Using a high-pass filter to extract the high frequencies of images. 2) Calculating the CC between the high frequencies to obtain the SCC. The commonly used Laplacian filter has the following form:

$$\mathbf{L} = \begin{bmatrix} -1 & -1 & -1 \\ -1 & 8 & -1 \\ -1 & -1 & -1 \end{bmatrix}. \tag{24}$$

The CC is another widely used spectral indicator which is defined as follows:

$$\text{CC} = \frac{\sum_{i=1}^{w} \sum_{j=1}^{h} (r_{ij} - \bar{r})(\hat{r}_{ij} - \bar{\hat{r}})}{\sqrt{\sum_{i=1}^{w} \sum_{j=1}^{h} (r_{ij} - \bar{r})^2 (\hat{r}_{ij} - \bar{\hat{r}})^2}}, \tag{25}$$

where $r_{ij}$ and $\hat{r}_{ij}$ are pixel values from GT and fused images, $\bar{r}$ and $\bar{\hat{r}}$ are their respective means, $w$ and $h$ are the width and height of the image.

(5) Quality Index (Q2$^n$): Q2$^n$ extends the universal image quality index (UIQI) metric to multi-spectral or hyper-spectral images and is computed as:

$$Q2^n = \frac{|\text{Cov}(\mathbf{r}, \hat{\mathbf{r}})|}{\sigma_\mathbf{r} \cdot \sigma_{\hat{\mathbf{r}}}} \cdot \frac{2\sigma_\mathbf{r} \cdot \sigma_{\hat{\mathbf{r}}}}{\sigma_\mathbf{r}^2 + \sigma_{\hat{\mathbf{r}}}^2} \cdot \frac{2|\bar{\mathbf{r}}| \cdot |\bar{\hat{\mathbf{r}}}|}{|\bar{\mathbf{r}}|^2 \cdot |\bar{\hat{\mathbf{r}}}|^2}, \tag{26}$$

where $\mathbf{r}$ and $\hat{\mathbf{r}}$ are multiband pixel vectors from the GT and fused image, respectively. $\text{Cov}(\cdot, \cdot)$, $\sigma_*$, and $\bar{*}$ denotes covariance, standard deviation and mean, respectively.

(6) Spectral Distortion Index (D$_\lambda$): This metric assesses spectral consistency among bands in the fused image compared to the low-resolution input by using inter-band Q values. Mathematically, it can be expressed as:

$$D_\lambda = \left( \frac{1}{B(B-1)} \sum_{i=1}^{B} \sum_{j=1(j\neq i)}^{B} |Q(\hat{r}_i, \hat{r}_j) - Q(y_i, y_j)|^q \right)^{\frac{1}{q}}, \tag{27}$$

where $Q(\cdot, \cdot)$ is the Quality Index between bands, $\hat{r}_i$ and $y_i$ denote the $i$-th band of the fused and low-resolution MS images, respectively. $q$ is typically set to 1.

(7) Spatial Distortion Index (D$_s$): D$_s$ measures spatial fidelity relative to the PAN image, which is complementary to $D_\lambda$. It is defined as follows:

$$D_s = \left( \frac{1}{B} \sum_{i=1}^{B} |Q(\hat{r}_i, p) - Q(y_i, \tilde{p})|^q \right)^{\frac{1}{q}}, \tag{28}$$

where $p$ is the original PAN image, $\tilde{p}$ represents its downsampled version, $Q(\cdot, \cdot)$ denotes the Quality Index and $q$ is typically set to 1.

(8) Hybrid Quality with No Reference (HQNR): HQNR provides an global no-reference quality estimate by combining spectral and spatial distortions ($D_\lambda$ and $D_s$). Specifically, it is defined as:

$$\text{HQNR} = (1 - D_\lambda^K)^\alpha \cdot (1 - D_s)^\beta, \tag{29}$$

where $K$, $\alpha$, and $\beta$ are hyperparameters (typically, $\alpha = \beta = 1$).

(9) Structural Similarity (SSIM): SSIM captures structural similarity by combining luminance and structural contrast functions:

$$\text{SSIM} = \frac{1}{B} \sum_{i=1}^{B} \frac{(2\mu_i \hat{\mu}_i + C_1)(2\sigma_{i\hat{i}} + C_2)}{(\mu_i^2 + \hat{\mu}_i^2 + C_1)(\sigma_i^2 + \hat{\sigma}_i^2 + C_2)}, \tag{30}$$

where $\mu_i, \hat{\mu}_i$ are the mean intensities of $I_i$ and $\hat{I}_i$, $\sigma_i^2, \hat{\sigma}_i^2$ are their variances, $\sigma_{i\hat{i}}$ is their covariance, $C_1$ and $C_2$ are fixed constants.

### A.3.3 Encoder Details: Bidirectional Transformer

The encoder is designed to jointly capture the spatial detail of the high-resolution PAN input and the spectral content of the coarser MS input. To this end, we adopt a bidirectional dilation transformer architecture with parallel spatial and spectral branches [12]. In the spatial branch, the PAN image is concatenated with the MS image after upsampling (bicubic interpolation to PAN resolution), then passed through an initial convolutional block that increases the channel dimensionality. This is followed by a cascade of Dilated Spatial Self-Attention (D-Spa) modules. Analogously, the spectral branch processes the original low-resolution MS image with its own convolutional block, followed by Grouped Spectral Self-Attention (G-Spe) modules. Each branch thus produces a hierarchy of feature maps at progressively deeper levels. By operating in a bidirectional, multi-scale fashion, these two branches extract complementary representations of the two inputs that can later be fused.

Specifically, we implement a dual-branch design: (i) a spatial branch applies 3 D-Spa layers with dilation rate 2 and 8 attention heads, and (ii) a spectral branch uses same 3 G-Spe layers and 8 attention heads with 8×8 non-overlapping spatial groups (i.e. window size is 8×8). Two branches operate in parallel and are fused to generate a unified latent representation for the following radiance estimation. Each D-Spa module implements spatial attention over a dilated neighborhood. Concretely,

the input feature map is linearly projected (via 1×1 convolutions or fully-connected layers) into query, key, and value tensors for each attention head. Instead of attending to a contiguous local window, D-Spa applies a fixed "hollow" window defined by a dilation rate. This dilation enlarges each token's receptive field without introducing additional parameters or sliding computations. In the spectral branch, each G-Spe module performs channel self-attention in a grouped manner over the spatial domain. The input feature map is first partitioned into a fixed number of spatial groups (i.e. non-overlapping patches). After processing all groups independently, their outputs are concatenated to reform the full feature map. This grouped design retains the dense channel connectivity and content-aware nature of spectral attention while focusing each operation on a local region, enabling the model to learn fine-grained spectral correlations within every area of the image.

Overall, the multi-level outputs from the D-Spa and G-Spe branches form a hierarchical representation, which is fused through feature integration and progressive upsampling to produce a unified high-resolution spectral-spatial representation. The enriched spectral-spatial feature map is then fed to inform the radiance estimation.

### A.3.4 Implementation and Training Details

The implementation of our proposed PINO uses PyTorch 2.1.0 and Python 3.10 on an Ubuntu 20.04.6 OS, with training conducted on an NVIDIA RTX 4090 GPU. A two-stage training strategy is adopted, with optimization performed using the Adam algorithm (with parameters $\beta_1 = 0.9$ and $\beta_2 = 0.999$) [31]. In the first stage, the encoder is trained independently using a learning rate of 0.0002 to minimize a composite loss function consisting of the MAE loss and the structural similarity index loss ($L_1 + \alpha_{s1}L_{ssim}$, where $\alpha_{s1} = 0.1$). The encoder is trained for 100, 600, and 2000 epochs on the WV3, GF2, and CAVE datasets, respectively. In the second stage, the entire framework is tuned with a learning rate halved from the first stage, optimizing a loss composed of MAE and histogram loss ($L_1 + \alpha_{s2}L_{hist}$, where $\alpha_{s2} = 0.01$). This stage is conducted over 200 epochs for WV3 and GF2, and 2000 epochs for CAVE.

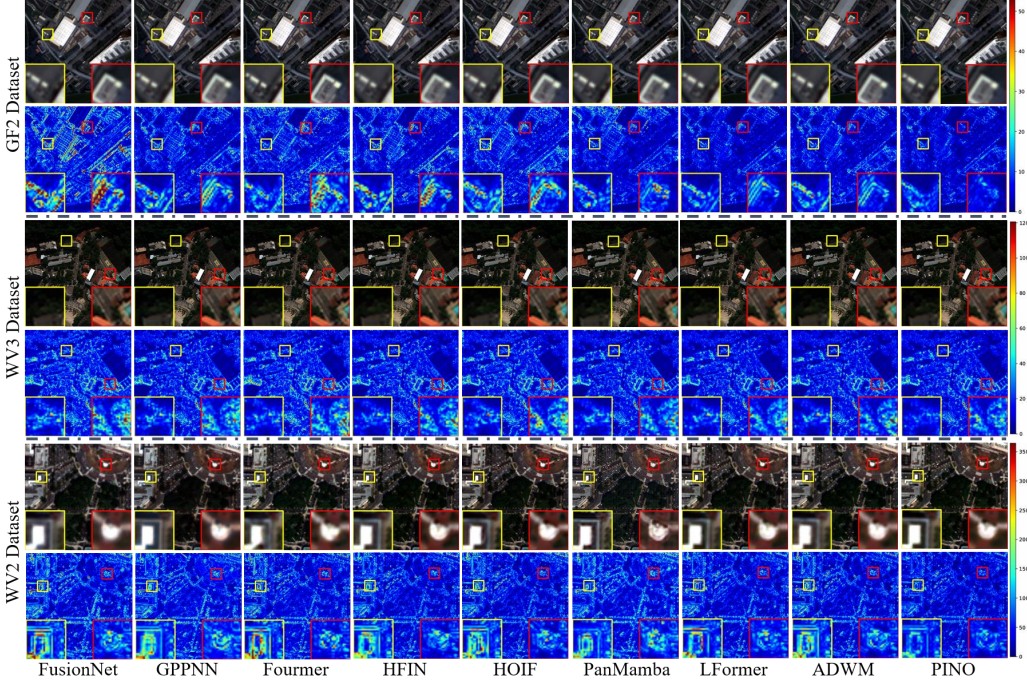

Figure 8: The visual results (odd rows) and the corresponding mean absolute error (MAE) maps (even rows) of all compared DL-based methods on reduced-resolution samples from the GF2, WV3 and WV2 sensors, respectively.

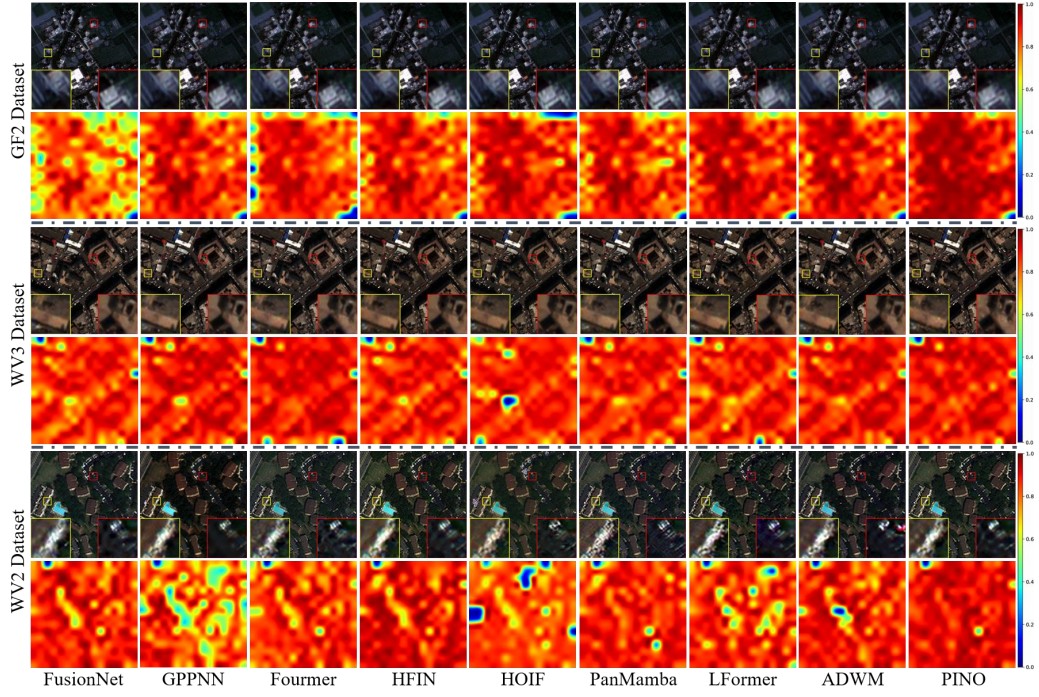

Figure 9: The visual results (odd rows) and the corresponding HQNR maps (even rows) of all compared DL-based methods on full-resolution samples from the GF2, WV3 and WV2 sensors, respectively.

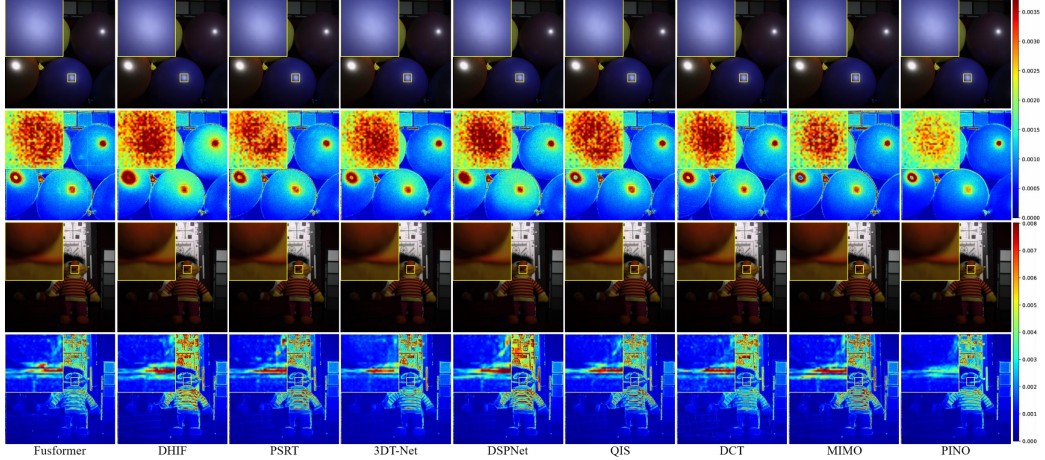

Figure 10: The visual results (odd rows) and the corresponding mean absolute error (MAE) maps (even rows) of all compared DL-based methods on two CAVE×4 testing samples.

### A.4 More Visualization Results on Pansharpening Benchmarks

To further illustrate the visual effectiveness of our proposed PINO framework, we provide additional pansharpening examples on reduced- and full-resolution samples from the WV3, GF2 and WV2 datasets. As shown in Figure 8 and 9, we compare PINO with all DL-based methods presented in the main paper using RGB visualization and corresponding error maps.

On reduced-resolution samples (Figure 8), our method exhibits clearer spatial textures and more faithful spectral appearance, especially in vegetation and urban structures. More importantly, the associated MAE maps reveal lower error distributions across most spatial regions, suggesting superior spectral-spatial consistency. On full-resolution benchmarks (Figure 9), PINO produces significantly sharper images with better preservation of high-frequency details, while achieving lower distortion in HQNR maps compared to competing baselines. These results confirm the robust performance of our method not only under supervised conditions but also in realistic, label-free scenarios.

### A.5 Visualization Results on Cave Dataset

We further evaluate the proposed framework on the CAVE hyperspectral dataset to verify its performance on multispectral-hyperspectral image fusion task. As shown in Figure 10, we visualize the fused results and corresponding error maps for two representative CAVE ×4 testing samples.

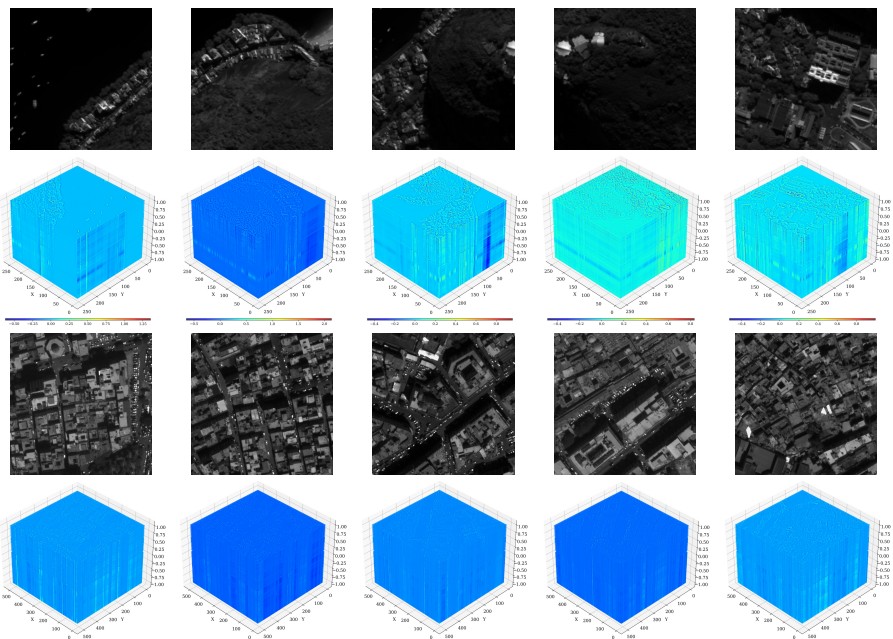

Figure 11: The visualized high-resolution PAN images (odd rows) and estimated radiance field (even rows) on 5 reduced-resolution samples and 5 full-resolution samples from the WV3 sensor, respectively. Please zoom in for more details.

Our method demonstrates high fidelity in reconstructing both spatial structures and spectral profiles, with MAE maps revealing minimal residuals across regions of varying complexity. Unlike competing methods, PINO effectively retains subtle spectral variations while preserving spatial continuity, particularly in edge-rich and texture-sensitive areas such as printed text, fabrics, and natural surfaces. These observations validate the versatility of our physics-informed design when extended to multispectral-hyperspectral image fusion task.

### A.6 Visualization of Learned Radiance Field and Spectral Responsivity

To qualitatively assess the spectral fidelity and physical interpretability of the proposed PINO framework, we visualize both the learned radiance field and the corresponding spectral responsivity functions. These visualizations aim to demonstrate how effectively the model captures the intrinsic characteristics of the spectral imaging process.

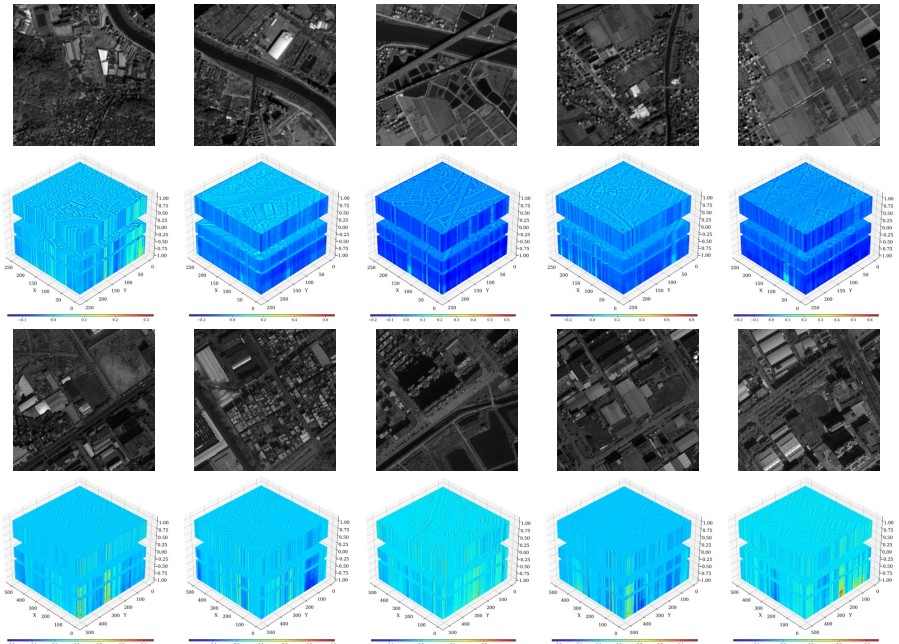

Figure 12: The visualized high-resolution PAN images (odd rows) and estimated radiance field (even rows) on 5 reduced-resolution samples and 5 full-resolution samples from the GF2 sensor, respectively. Please zoom in for more details.

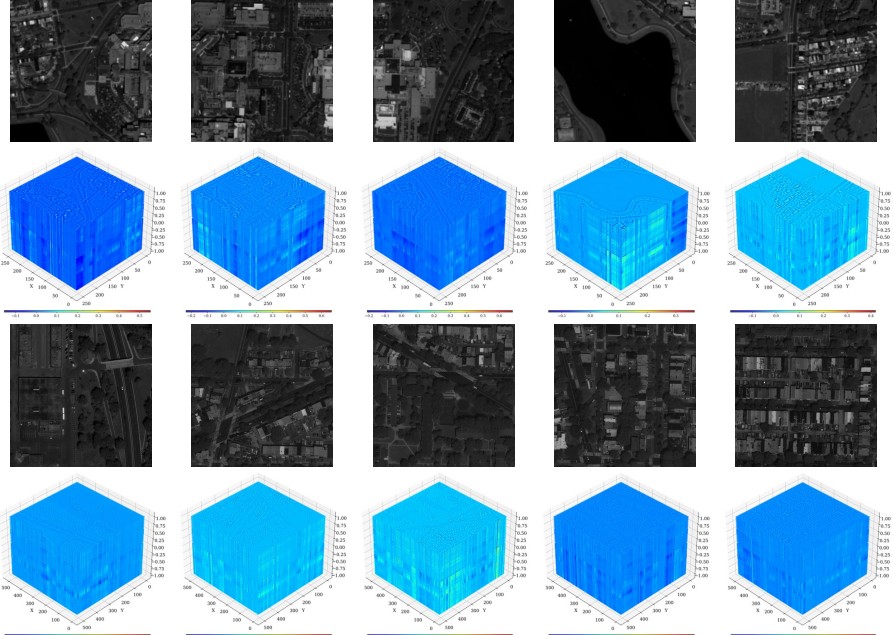

Figure 13: The visualized high-resolution PAN images (odd rows) and estimated radiance field (even rows) on 5 reduced-resolution samples and 5 full-resolution samples from the WV2 sensor, respectively. Please zoom in for more details.

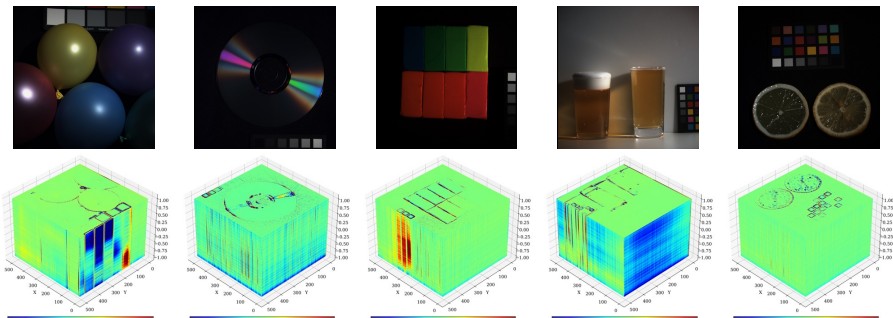

Figure 14: The visualized high-resolution RGB images (first row) and estimated radiance field (second row) on 5 test samples from the CAVE dataset, respectively. Please zoom in for more details.

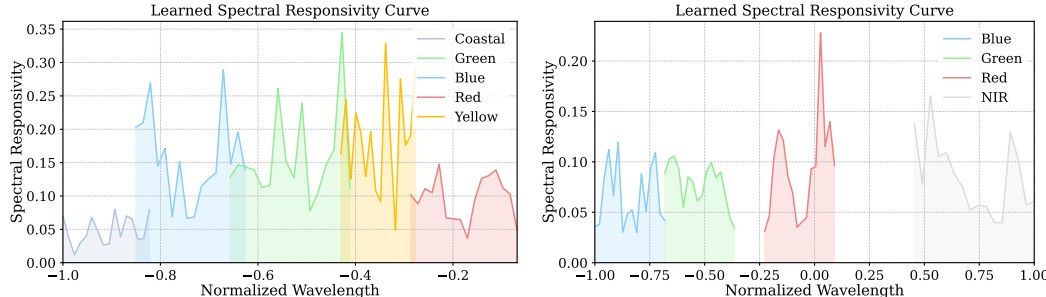

Figure 15: The learned spectral responsivity curves across various spectral bands and different datasets. Left: five bands from the WV3 dataset; Right: four bands from the GF2 dataset.

Figures 11, 12, 13, and 14 present visualizations of the learned radiance field on both reduced- and full-resolution test samples from the WV3, GF2, WV2 datasets, and test samples from CAVE x4 dataset. The radiance field is rendered based on the indexed spatial coordinates and wavelengths (randomly sampled). Compared to the input PAN image, the learned radiance map preserves fine spatial details while modulating them with wavelength-dependent structures. This highlights the capacity of PINO to simultaneously resolve high-frequency textures and continuous spectral variation.

Complementing this, Figure 15 illustrates the learned spectral responsivity curves on the WV3 and GF2 datasets (for better visualization, we select five bands for WV3). Each curve represents the model's learned responsivity over the wavelength domain for a specific sensor band. Notably, the curves exhibit band-specific shapes similar to real sensor characteristics. This indicates that the model successfully internalizes the spectral selectivity imposed by sensor filters, despite learning these responsivity functions purely from data, without explicit supervision.

Together, these results validate the model's capacity to approximate the underlying physical formation of sensor measurements through implicit neural modeling. The radiance field captures the latent spectral-spatial structure of the scene, while the learned responsivity functions align closely with real-world sensor profiles. Such physically grounded representations offer both enhanced interpretability and robust generalization across varying sensor domains.

