# OpenReview forum: "Physics-informed Neural Operator for Pansharpening"
_NeurIPS.cc/2025/Conference — NeurIPS 2025 poster_

### Official Review · Reviewer_eGPc · 2025-06-29

**Clarity:** 3
**Significance:** 3
**Originality:** 3
**Rating:** 4
**Confidence:** 4

**Summary:**

This paper proposes a novel physics-informed neural operator framework for pansharpening, termed PINO. A spatial-spectral encoder is introduced to aggregate multi-granularity high-resolution panchromatic (PAN) and low-resolution multi-spectral (LRMS) features. An iterative neural integral process utilizes these fused spatial-spectral characteristics to learn a continuous radiance field. The learned radiance field is modulated by the sensor’s spectral responsivity to produce the desired fusion products.

**Questions:**

See the weakness.

**Ethical Concerns:**

["NO or VERY MINOR ethics concerns only"]

**Final Justification:**

The paper investigates the physical imaging process and corresponding mathematical modeling, which serve as the theoretical foundation of the proposed approach.

**Quality:**

3

**Strengths And Weaknesses:**

Strengths
1, this paper proposed a novel physics-informed neural operator framework for pansharpening, termed PINO.
2, A spatial-spectral encoder is introduced to aggregate multi-granularity high-resolution panchromatic (PAN) and low-resolution multi-spectral (LRMS) features.
3, An iterative neural integral process utilizes these fused spatial-spectral characteristics to learn a continuous radiance field.
Weaknesses
1, It would be better to describe section 3.1 in the background section.
2, It seems that the contribution of this paper comes from the utilization of PIOO for Pansharpening. Theoretical analysis is encouraged to investigate.

---

> ### Author Rebuttal · Authors · 2025-07-31
>
> Dear Reviewer,
>
> We thank you for your careful comments on our paper and sincerely appreciate your recognition of our work as “a novel physics-informed neural operator framework.”
>
> We will carefully revise the paper in accordance with your comments. Below, we provide point-by-point responses to the weaknesses (W).
>
> **W1. Reorganization of Sec. 3.1:**
>
> Thank you for this instructive suggestion, we will move Sec. 3.1 to a new “Background and Motivation” section for better clarity.
>
> **W2. Contributions and Theoretical analysis:** We would like to clarify that the primary contribution of our paper lies in the theoretical analysis of the physical imaging process and mathematical modeling, which leads to a physically principled solution applicable to both multispectral and hyperspectral image fusion tasks.
>
> **Rather than simply introducing a neural operator as a black-box component, our method leverages the iterative kernel integral operator as a functional approximator for the spectral integral in Eq. 2, allowing the learning of a continuous radiance field modulated by spatial coordinates and wavelength**. This design is grounded in the theoretical analysis of the physical sensor imaging process, as detailed in Section 3.2. It enables sub-pixel and sub-band fusion of different modalities, which is essential for the pansharpening task. Although the ablation studies (please see Sec 4.2 in the main paper) demonstrate that the neural operator is critical to the performance improvement, the wavelength modulation brings more benefits to our final result.
>
> Additionally, our framework incorporates learnable band-wise spectral responsivity modulation to simulate the sensor’s spectral properties. This component is also crucial to the improvement in performance. We provide visualizations of the learned radiance field and spectral responsivity in the appendix A.4, which further support the interpretability and effectiveness of the proposed method and its components. We believe these contributions offer a solid theoretical foundation for advancing the field of pansharpening. The mentioned contributions are summarized in our main paper, and also presented in the review summary.
>
> To further clarify the role of the iterative kernel integral operator, we provide the following theoretical analysis of our approach:
>
> Our approach is grounded in a detailed formulation of the physical imaging process and corresponding mathematical modeling, which serve as the theoretical foundation of our approach. To clarify this further, we revisit and summarize the key components of our method as follows:
>
> **1. Sensor Physics Modeling:**
>
> (1) Radiance formation follows the camera equation (Eq. 1 in our main paper): $L_i(x, y, \lambda)=\frac{\pi \tau_o(\lambda)}{4 N^2} L_i^{s}(x, y, \lambda)$, which is implicitly realized through a spatial-spectral encoder within our PINO.
>
> (2) Spectral integration (Eq. 2 in our main paper) models the sensor's band-wise response: $I_{i, b}(x, y) = \int_{\Lambda_b} L_i(x, y, \lambda)R_b(\lambda)\$. In our PINO, the spectral integration is approximated via an iterative kernel integral (*i.e.*, neural operator) in Eq. 8, while the sensor responsivity $R_b(\lambda)$ is implemented as a learnable function, modulated using Eq. 9.
>
> (3) Discretization (Eq. 3 in our main paper) enables the practical implementation of the integration, reflecting the quantization of real sensor outputs: $I_{i,b}(x,y)\approx\sum_j L_i(x,y,\lambda_j)R_b(\lambda_j)\Delta\lambda_j=\sum_j L_i(x,y,\lambda_j)R^{\prime}_b(\lambda_j)$.
>
> This formulation explicitly encodes key electro-optical processes into the learning framework, ensuring that PINO is physically consistent and spectrally faithful.
>
> **2. Neural Operator as Physics-Informed Solver:**
>
> The iterative Galerkin-style kernel integral operator (Eq.8 in our main paper) is designed as a functional approximator for the integral in Eq. 2: $h^{n+1}(x, y, \lambda) = \int_{\Lambda} \left( \int_{\Omega_x} \int_{\Omega_y} \mathcal{K} \left( (x, y), (u, v) \right) h^{n}(u, v) \, \mathrm{d}u \, \mathrm{d}v \right) \Pi(\lambda) \mathrm{d}\lambda'$, where $\mathcal{K}$ captures spatial-radiance coupling;  $\Pi(\lambda)$ implements implicit spectral basis functions via Fourier encoding; discretization (*i.e.*, wavelength modulation) mirrors sensor quantization. This iterative integral allows the model to learn a continuous radiance field that aligns with the physical principles of radiative imaging.
>
> **3. Spectral Responsivity as Differentiable Physics:**
>
> To account for sensor-specific spectral filtering, we parameterize each band’s responsivity $R^\prime_b(\lambda_j, \theta_b)$ as: $R^\prime_b(\lambda_j, \theta_b) = \mathrm{Sigmoid}(\theta_b(\Pi(\lambda_j)))$. In this formulation, $R^\prime_b(\lambda_j, \theta_b)\in [0, 1]$ guarantees the non-negativity that aligns with the bounded nature of real sensor spectral responses; the differentiable sigmoid function (with gradient $\sigma'(x) = \sigma(x) (1 - \sigma(x))$) enables smoothness, avoiding non-physical spectral discontinuities; the learnable MLP mapping $\theta_b(\cdot)$ allows the model to adaptively fit sensor-specific band responses from data.
>
> We believe that this deeper theoretical insight further clarifies how the neural operator, wavelength modulation, and learnable spectral responsivity collectively contribute to the accuracy and effectiveness of our pansharpening framework. The extra content about more detailed theoretical analysis will be included in the revised version of the Appendix for better clarity.

---

> > ### Comment · Reviewer_eGPc · 2025-08-03
> >
> > Thanks for the responses. My concerns are addressed. I have raised my scores.

---

> ### Author Response · Authors · 2025-08-04
> **Appreciation for the Reviewer’s Acknowledgment of the Rebuttal**
>
> Dear Reviewer,
>
> We sincerely appreciate the time and effort you have dedicated to reviewing our manuscript, as well as your positive recognition of our rebuttal.

---

### Official Review · Reviewer_qiJf · 2025-07-01

**Clarity:** 2
**Significance:** 2
**Originality:** 3
**Rating:** 5
**Confidence:** 4

**Summary:**

This paper presents an interesting model called PINO for pansharpening, which combines deep learning with the underlying physics of spectral imaging. The approach is implemented using implicit neural representations and neural operators, offering a novel perspective on the problem. Given that physics-driven methods are still relatively rare in remote sensing (RS) research, this work could inspire more explorations in this direction. Overall, the paper is well-structured, and the experimental results are compelling.

**Questions:**

See the Weaknesses.

**Ethical Concerns:**

["NO or VERY MINOR ethics concerns only"]

**Final Justification:**

Thanks for the responses. My concerns are addressed. I'll maintain my original rating.

**Limitations:**

Yes.

**Paper Formatting Concerns:**

No major formatting issues.

**Quality:**

3

**Strengths And Weaknesses:**

# Strengths
1. The paper introduces an interesting method for pansharpening that integrates physical imaging using implicit neural representation and neural operators. The idea is innovative compared with existing pansharpening works.
2. The experiment results are significantly improved over the existing methods, while the ablation studies show the effectiveness of the core designs.
3. The motivation and presentation of the paper are clear.

# Weaknesses
1. While combining pansharpening with physical imaging is reasonable, the paper needs deeper theoretical analysis of the proposed model, particularly regarding its physical modeling components.
2. The physical interpretability of the proposed model remains unclear. Additional experiments or in-depth analysis would help demonstrate whether it properly captures physical imaging properties.
3. The discussion should include more relevant physically-driven pansharpening works, such as those presented in DOI: 10.1109/TGRS.2024.3523865, for better contextualization.

---

> ### Author Rebuttal · Authors · 2025-07-31
>
> Dear Reviewer,
>
> We sincerely appreciate your detailed feedback and insightful comments, which are invaluable for enhancing the quality of our paper. We are especially encouraged by the comments: “the idea is innovative compared with existing pansharpening works, the experiment results are significantly improved, the motivation and presentation of the paper are clear.”
>
> We will carefully revise the paper in accordance with your comments. Below, we provide point-by-point responses to the weaknesses (W).
>
> **W1. Theoretical analysis of physical modeling:**
> Our approach is grounded in a detailed formulation of the physical imaging process and corresponding mathematical modeling, which serve as the theoretical foundation of our approach. To clarify this further, we revisit and summarize the key components of our method as follows:
>
> **1. Sensor Physics Modeling:**
>
> (1) Radiance formation follows the camera equation (Eq. 1 in our main paper): $L_i(x, y, \lambda)=\frac{\pi \tau_o(\lambda)}{4 N^2} L_i^{s}(x, y, \lambda)$, which is implicitly realized through a spatial-spectral encoder within our PINO.
>
> (2) Spectral integration (Eq. 2 in our main paper) models the sensor's band-wise response: $I_{i, b}(x, y) = \int_{\Lambda_b} L_i(x, y, \lambda)R_b(\lambda)\$. In our PINO, the spectral integration is approximated via an iterative kernel integral (*i.e.*, neural operator) in Eq. 8, while the sensor responsivity $R_b(\lambda)$ is implemented as a learnable function, modulated using Eq. 9.
>
> (3) Discretization (Eq. 3 in our main paper) enables the practical implementation of the integration, reflecting the quantization of real sensor outputs: $I_{i,b}(x,y)\approx\sum_j L_i(x,y,\lambda_j)R_b(\lambda_j)\Delta\lambda_j=\sum_j L_i(x,y,\lambda_j)R^{\prime}_b(\lambda_j)$.
>
> The above formulas explicitly encodes key electro-optical process into the learning framework, ensuring that PINO is physically consistent and spectrally faithful.
>
> **2. Neural Operator as Physics-Informed Solver:**
>
> The iterative Galerkin-style kernel integral operator (Eq. 8 in our main paper) is designed as a functional approximator for the integral in Eq. 2 of the mian paper: $h^{n+1}(x, y, \lambda) = \int_{\Lambda} \left( \int_{\Omega_x} \int_{\Omega_y} \mathcal{K} \left( (x, y), (u, v) \right) h^{n}(u, v) \, \mathrm{d}u \, \mathrm{d}v \right) \Pi(\lambda) \mathrm{d}\lambda'$, where $\mathcal{K}$ captures spatial-radiance coupling;  $\Pi(\lambda)$ implements implicit spectral basis functions via Fourier encoding; discretization (*i.e.*, wavelength modulation) mirrors sensor quantization. This iterative integral allows the model to learn a continuous radiance field that aligns with the physical principles of radiative imaging.
>
> **3. Spectral Responsivity as Differentiable Physics:**
>
> To account for sensor-specific spectral filtering, we parameterize each band’s responsivity $R^\prime_b(\lambda_j, \theta_b)$ as: $R^\prime_b(\lambda_j, \theta_b) = \mathrm{Sigmoid}(\theta_b(\Pi(\lambda_j)))$. In this formulation, $R^\prime_b(\lambda_j, \theta_b)\in [0, 1]$ guarantees the non-negativity that aligns with the bounded nature of real sensor spectral responses; the differentiable sigmoid function (with gradient $\sigma'(x) = \sigma(x) (1 - \sigma(x))$) enables smoothness, avoiding non-physical spectral discontinuities; the learnable MLP mapping $\theta_b(\cdot)$ allows the model to adaptively fit sensor-specific band responses from data.
>
> We believe that this deeper theoretical insight further clarifies how the neural operator, wavelength modulation, and learnable spectral responsivity collectively contribute to the accuracy and effectiveness of our pansharpening framework. The extra content about more detailed theoretical analysis will be included in the revised version of the Appendix for better clarity.
>
> **W2. Physical interpretability:**
>
> 1. We would like to emphasize that the proposed model is physically interpretable by design, as it explicitly incorporates a wavelength-modulated radiance field and a learnable spectral responsivity function. This modeling aligns closely with the physical principles of sensor imaging physics, as detailed in our preceding analysis.
>
> 2. Moreover, we visualize the learned radiance maps and sensor responsivity curves in Appendix A.4. The radiance maps preserve fine-grained structures and exhibit wavelength-dependent variations, while the learned responsivity curves resemble real-world sensor spectral responses. These results support the model’s ability to capture and reproduce key physical characteristics of the imaging process, providing further evidence of interpretability.
>
> To further improve clarity, we will add a related description of these analyses in the main paper as per your instructive suggestion.
>
> **W3. More discussion about relevant physically-driven pansharpening works:**
>
> We thank the reviewer for pointing this out and for suggesting relevant references. In the revised version, we will update the Related Work section to better contextualize our approach by including recent physically-driven pansharpening methods, such as the SSMNet and FBS-PS [R1, R2], both of which incorporate spectral-band separability based on sensor characteristics to inform network design. While these prior works has introduced model components inspired by data properties or spectral decomposition, our method uniquely formalizes the imaging process via a learnable wavelength-modulated radiance field and band-specific sensor response modeling, thereby offering a principled physically-informed framework.
>
> [R1] Kim H H, Kim M. FBS-PS: Fully Band-Separable PAN-Sharpening Considering the Physical Characteristics of Electro-Optical Sensors[J]. IEEE Transactions on Geoscience and Remote Sensing, 2024.
>
> [R2] Liu X, Hou J, Cong X, et al. Rethinking pan-sharpening via spectral-band modulation[J]. IEEE Transactions on Geoscience and Remote Sensing, 2023, 62: 1-16.

---

> > ### Comment · Reviewer_qiJf · 2025-08-09
> > **Official Comment by Reviewer**
> >
> > I appreciate the authors' thoughtful response, which has addressed my initial concerns. I maintain my original score of 5 as the rebuttal has confirmed the technical soundness and novelty of this work.

---

> ### Author Response · Authors · 2025-08-04
> **Appreciation for the Reviewer’s Acknowledgment of the Rebuttal**
>
> Dear Reviewer,
>
> We sincerely appreciate your valuable comments and the time you have dedicated to reviewing our manuscript. If you have any additional concerns that remain unaddressed, please feel free to let us know at any time. We would be pleased to provide additional clarification.

---

### Official Review · Reviewer_8nfB · 2025-07-02

**Clarity:** 3
**Significance:** 2
**Originality:** 3
**Rating:** 4
**Confidence:** 4

**Summary:**

This paper proposed a physics-informed neural operator framework, named PINO, which unified continuous radiance field modeling with learnable spectral responsivity, significantly boosting spatial spectral fidelity and enhancing generalization across heterogeneous sensors and varying imaging conditions.
The main contributions as follows:
1)	This paper proposed a physically-grounded pansharpening framework that established a new paradigm bridging neural representations with physical sensor model.
2)	It employed an iterative kernel integral operator that leverages multi-granularity spatial–spectral features to learn a continuous radiance field over spatial coordinates and wavelength, enabling sub-pixel and sub-band fusion of different modalities while effectively emulating band-wise spectral integration.
3)	Introduced a learnable band-wise spectral responsivity modulation to simulate the sensor’s spectral properties, allowing the simultaneous optimization of response functions.
4)	Experiments on multiple remote sensing benchmark datasets reveal that PINO consistently outperforms state-of-the-art methods in fusion capability, generaizalibity and adaptability.

**Questions:**

1.	Why implement analog imaging process operators in imaging products? From a physical principle perspective, why is this useful?
2.	Lack of more detailed experimental analysis.
3.	Will the approach used in this paper result in a significant increase in computational load and a significant decrease in training speed?

**Ethical Concerns:**

["NO or VERY MINOR ethics concerns only"]

**Final Justification:**

Thanks for the responses.. I'll maintain my original rating.

**Limitations:**

yes

**Quality:**

3

**Strengths And Weaknesses:**

Strengths :
The idea is novel, and the theoretical reasoning is correct.
Weaknesses:
1.Readers are not clear about why physical information neural operators are useful.
2. The experimental data set is small and insufficient.

---

> ### Author Rebuttal · Authors · 2025-07-31
>
> Dear Reviewer,
>
> We sincerely appreciate your detailed feedback and instructive comments, which are invaluable for enhancing the quality of our paper. We are especially encouraged by the comment: “The idea is novel, and the theoretical reasoning is correct.”
>
> We will carefully revise the paper in accordance with your comments. Below, we provide point-by-point responses to your questions (Q) and identified weaknesses (W).
>
> **W1 & Q1. Why implement analog imaging process operators in imaging products? From a physical principle perspective, why is this useful?**
>
> To clarify, our goal is not to explicitly implement an analog imaging process, but rather to design learning components inspired by the underlying *physical principles and mathematical models* that govern the sensor imaging process. Specifically, pansharpening requires fusing low-spatial high-spectral MS images with high-spatial low-spectral PAN (or RGB) images to reconstruct high-resolution multispectral images. This inherently involves the inverse of a physical image formation process that is both spatially and spectrally entangled. By analyzing the mathematical formulation behind this process (rather than relying purely on end-to-end black-box learning), we can guide network design to more effectively leverage complementary modalities and enhance interpretability.
>
> This paradigm is conceptually aligned with advances in other areas such as NeRF, 3D Gaussian Splatting, and Physics-Informed Neural Networks [R1-R3], where physical models inform the training pipeline or network architectures. While prior work in pansharpening has introduced model components inspired by data properties [R4, R5], our approach uniquely formalizes the imaging process through a learnable wavelength-modulated radiance field and band-specific sensor response modeling. These design choices not only yield superior performance but also enhance interpretability. As shown in Appendix A.4, our model learns radiance maps that preserve fine-grained structures and wavelength-dependent variations, as well as sensor responsivity curves that resemble real-world spectral responses, further supporting the value of this physically-informed modeling approach.
>
> **We kindly refer you to our response to Q1 from Reviewer qiJf, where we provide a detailed explanation of the sensor imaging physics and how it is explicitly implemented in our proposed method.**
>
> **W2 & Q2. Experimental data and more detailed experimental analysis:** We respectfully clarify that our experimental setup follows widely accepted protocols in the pansharpening community [R6-R8]. We evaluate our model on three standard satellite datasets: WV3, WV2, and GF2, with training samples simulated from original satellite imagery using Wald’s protocol [R9]. We report results under both reduced (256×256) and full (512×512) resolution settings, and importantly, we also evaluate cross-dataset generalization on GF2, which is not seen during training. Beyond pansharpening, we validate spectral learning capacity on hyperspectral datasets (CAVE ×4 and CAVE ×8), demonstrating the model’s ability to generalize to diverse spectral settings.
>
> To complement quantitative metrics, we provide detailed qualitative analysis in the original paper and appendix, including RGB outputs, HQNR maps, and comparison visualisations across benchmarks. The results show that PINO effectively retains subtle spectral variations while preserving spatial continuity, especially in edge-rich or texture-sensitive regions. Furthermore, we visualize internal representations such as the learned radiance field and spectral responsivity, which confirm the model’s ability to reconstruct fine spatial structures while learning band-specific spectral selectivity imposed by the sensor. These analyses support both the performance and interpretability of our approach.
>
> **Q3. Computational load and training speed:**
>
> While we agree that the introduction of iterative neural integration and radiance field modulation does result in increased training time in some cases, we would like to clarify that our two-stage training approach reduces training time in certain scenarios.
>
> To be specific, we summarize the training epochs and inference speed (computational load) as follows:
>
> (1) **WV3 dataset**: The encoder is trained for 100 epochs, followed by 200 epochs in the second stage. In comparison, the baseline methods such as LFormer requires 800 epochs, and Fourmer, ADWM and PanMamba require 500 epochs [R10-R13].
> Thus, the total number of training epochs of our method is reduced compared to the baseline methods.
>
> For a fair comparison, we evaluate the inference speed of our method and several recent baselines on the WV3 dataset with a resolution of 256×256:
>
> Ours: 0.064 s;   Fourmer: 0.263 s;   LFormer: 0.233 s;   ADWM: 0.022 s;   PanMamba: 0.034 s
>
> It is clear that our method delivers competitive inference speed while achieving state-of-the-art performance.
>
> (2) **GF2 dataset**: The encoder is trained for 600 epochs, followed by 200 epochs in the second stage. In comparison, the baseline methods such as LFormer requires 800 epochs, and Fourmer, ADWM and PanMamba require 500 epochs [R10-R13].
> Thus, the total training epochs of our approach are competitive with these baselines.
>
> (3) **CAVE dataset**: While it does indeed require 2000 epochs, it is important to note that baseline methods such as BDT and PSRT also require 2000 epochs [R14, R15], and QIS-GAN requires 10,000 training epochs [R16]. Thus, our method does not significantly increase the training time compared to these representative approaches.
>
> [R1] Kerbl B, Kopanas G, Leimkühler T, et al. 3D Gaussian splatting for real-time radiance field rendering[J]. ACM Trans. Graph., 2023, 42(4): 139:1-139:14.
>
> [R2] Mildenhall B, Srinivasan P P, Tancik M, et al. Nerf: Representing scenes as neural radiance fields for view synthesis[J]. Communications of the ACM, 2021, 65(1): 99-106.
>
> [R3] Raissi M, Perdikaris P, Karniadakis G E. Physics-informed neural networks: A deep learning framework for solving forward and inverse problems involving nonlinear partial differential equations[J]. Journal of Computational physics, 2019, 378: 686-707.
>
> [R4] Kim H H, Kim M. FBS-PS: Fully Band-Separable PAN-Sharpening Considering the Physical Characteristics of Electro-Optical Sensors[J]. IEEE Transactions on Geoscience and Remote Sensing, 2024.
>
> [R5] Liu X, Hou J, Cong X, et al. Rethinking pan-sharpening via spectral-band modulation[J]. IEEE Transactions on Geoscience and Remote Sensing, 2023, 62: 1-16.
>
> [R6] Hou J, Cao Q, Ran R, et al. Bidomain modeling paradigm for pansharpening[C]//Proceedings of the 31st ACM international conference on multimedia. 2023: 347-357.
>
> [R7] Vivone G, Deng L J, Deng S, et al. Deep learning in remote sensing image fusion: Methods, protocols, data, and future perspectives[J]. IEEE Geoscience and Remote Sensing Magazine, 2024.
>
> [R8] Zhong Y, Wu X, Cao Z, et al. Ssdiff: Spatial-spectral integrated diffusion model for remote sensing pansharpening[J]. Advances in Neural Information Processing Systems, 2024, 37: 77962-77986.
>
> [R9] Wald L, Ranchin T, Mangolini M. Fusion of satellite images of different spatial resolutions: Assessing the quality of resulting images[J]. Photogrammetric engineering and remote sensing, 1997, 63(6): 691-699.
>
> [R10] Zhou M, Huang J, Guo C L, et al. Fourmer: An efficient global modeling paradigm for image restoration[C]//International conference on machine learning. PMLR, 2023: 42589-42601.
>
> [R11] Hou J, Cao Z, Zheng N, et al. Linearly-evolved transformer for pan-sharpening[C]//Proceedings of the 32nd ACM international conference on multimedia. 2024: 1486-1494.
>
> [R12] Huang J, Chen H, Ren J, et al. A General Adaptive Dual-level Weighting Mechanism for Remote Sensing Pansharpening[C]//Proceedings of the Computer Vision and Pattern Recognition Conference. 2025: 7447-7456.
>
> [R13] He X, Cao K, Zhang J, et al. Pan-mamba: Effective pan-sharpening with state space model[J]. Information Fusion, 2025, 115: 102779.
>
> [R14] Deng S, Deng L J, Wu X, et al. Bidirectional dilation transformer for multispectral and hyperspectral image fusion[C]//Proceedings of the Thirty-Second International Joint Conference on Artificial Intelligence. 2023: 3633-3641.
>
> [R15] Deng S Q, Deng L J, Wu X, et al. PSRT: Pyramid shuffle-and-reshuffle transformer for multispectral and hyperspectral image fusion[J]. IEEE Transactions on Geoscience and Remote Sensing, 2023, 61: 1-15.
>
> [R16] Zhu C, Deng S, Zhou Y, et al. QIS-GAN: A lightweight adversarial network with quadtree implicit sampling for multispectral and hyperspectral image fusion[J]. IEEE Transactions on Geoscience and Remote Sensing, 2023, 61: 1-15.

---

> ### Author Response · Authors · 2025-08-04
> **Appreciation for the Reviewer’s Acknowledgment of the Rebuttal**
>
> Dear Reviewer,
>
> We sincerely appreciate your valuable comments and the time you have dedicated to reviewing our manuscript. If you have any further questions or concerns that remain unaddressed, please feel free to let us know at any time. We would be pleased to provide additional clarification.

---

### Official Review · Reviewer_3YGA · 2025-07-03

**Clarity:** 3
**Significance:** 3
**Originality:** 3
**Rating:** 4
**Confidence:** 4

**Summary:**

The PINO paper proposes a physics-based neural operator framework for pansharpening of remote sensing images. By simulating the physical imaging process, this framework provides a new solution that can significantly improve the quality and generalization ability of fused images. Although there are some challenges in terms of computational complexity and model complexity, PINO's outstanding performance on multiple benchmark datasets demonstrates its potential and value in the field of pansharpening. Future research can further optimize computational efficiency, explore cross-sensor generalization capabilities, and expand its application in other image fusion tasks.

**Questions:**

1. The encoder details are not detailed enough. The BDT architecture does not specify key parameters such as the number of layers/heads, and the appendix only mentions "Bidirectional Dilated Transformer".
2. For the boundary conditions of the radiation field, Formula 1 does not discuss the boundary constraints of the scene radiation $L_i^s$, which may affect physical consistency.
3. In the implementation of wavelength modulation, the discretization does not specify the sampling strategy (uniform/adaptive?).

**Ethical Concerns:**

["NO or VERY MINOR ethics concerns only"]

**Final Justification:**

The author has solved my questions in detail, so I will increase my score.

**Limitations:**

Same weaknesses and problems as before.

**Paper Formatting Concerns:**

Overall, the paper is grammatically clear, with no obvious grammatical errors. The author's language is accurate and logically coherent when describing the methods and experimental results.

**Quality:**

3

**Strengths And Weaknesses:**

# Strengths
1. The continuous radiation field modeling $L_i(x,y,\lambda)$ is combined with the learnable spectral response modulation $R_b(\lambda)$ to explicitly simulate the physical process of sensor imaging, surpassing the traditional heuristic fusion method. The learned spectral response curve is consistent with the real sensor characteristics, enhancing the credibility of the model.

2. Continuous radiation field modeling is achieved through Galerkin-type integration, preserving sub-pixel spatial details and sub-band spectral features. Pre-train the encoder (L1+SSIM) and then fine-tune the whole (L1+histogram loss) to alleviate the information degradation problem in implicit representation training.

3. Multi-granularity feature aggregation (expanded spatial attention + grouped spectral attention) effectively fuses the HR details of PAN and the spectral information of LRMS.

# Weaknesses
1. The high computational cost caused by iterative neural integration and radiation field modulation makes training time-consuming (2000 rounds for CAVE), and the authors did not evaluate the inference speed.

---

> ### Author Rebuttal · Authors · 2025-07-31
>
> Dear Reviewer,
>
> We sincerely appreciate your thorough review and insightful suggestions, which are invaluable for enhancing the quality of our paper. We are especially encouraged by the comments: “this framework provides a new solution, surpassing traditional heuristic fusion methods, with outstanding performance.”
>
> We will carefully revise the paper in accordance with your feedback. Below, we provide point-by-point responses to your questions (Q) and identified weaknesses (W).
>
> **W1. Number of training epochs and inference speed:**
> While we agree that the introduction of iterative neural integration and radiance field modulation does result in increased training time in some cases, we would like to clarify that our two-stage training approach reduces training time in certain scenarios.
>
> To be specific, we summarize the training epochs and inference speed as follows:
>
> (1) **WV3 dataset**: The encoder is trained for 100 epochs, followed by 200 epochs in the second stage. In comparison, the baseline methods such as LFormer requires 800 epochs, and Fourmer, ADWM and PanMamba require 500 epochs [R1-R4].
> Thus, the total number of training epochs of our method is reduced compared to the baseline methods.
>
> For a fair comparison, we evaluate the inference speed of our method and these recent baselines on the WV3 dataset with a resolution of 256×256:
>
> Ours: 0.064 s;   Fourmer: 0.263 s;   LFormer: 0.233 s;   ADWM: 0.022 s;   PanMamba: 0.034 s
>
> It is clear that our method delivers competitive inference speed while achieving state-of-the-art performance.
>
> (2) **GF2 dataset**: The encoder is trained for 600 epochs, followed by 200 epochs in the second stage. In comparison, the baseline methods such as LFormer requires 800 epochs, and Fourmer, ADWM and PanMamba require 500 epochs [R1-R4].
> Thus, the total training epochs of our approach are competitive with these baselines.
>
> (3) **CAVE dataset**: While it does indeed require 2000 epochs, it is important to note that baseline methods such as BDT and PSRT also require 2000 epochs [R5, R6], and QIS-GAN requires 10,000 training epochs [R7]. Thus, our method does not significantly increase the training time compared to these representative approaches.
>
> Additionally, the number of training epochs used in our experiments follows the original BDT method, which we do not adjust manually. Our primary focus in this work is to propose a physics-informed modeling approach for pansharpening that outperforms existing methods, rather than optimizing computational cost or complexity. The number of training epochs is, to some extent, a hyperparameter and is dependent on factors such as dataset size, learning rate, batch size, and pixel sampling per image. In future work, we could further optimize these hyper-parameters to reduce computational overhead.
>
> **Q1. Encoder architectural details:**
> Our encoder is based on the Bidirectional Dilation Transformer introduced in [R5] and we have presented details about the encoder input, up-sampling operation, core modules, feature embedding process and its design principle in the Appendix A.1.3. Specifically, we implement a dual-branch design:
> (i) a spatial branch applies $3$ Dilated Spatial Self-Attention (D-Spa) layers with dilation rate $2$ and $8$ attention heads,
> and (ii) a spectral branch uses same $3$ Grouped Spectral Self-Attention (G-Spe) layers and $8$ attention heads with 8$\times$8 non-overlapping spatial groups (i.e. window size is 8$\times$8).
>
> These details will be added to the appendix, and full code/configuration will be released upon acceptance for reproducibility.
>
> **Q2. Boundary conditions of the radiance field $\( L_i^s(x, y, \lambda) \) $ and physical consistency:** Thank you for your insightful comment on the boundary conditions of the radiation field in Formula 1. Although we did not explicitly discuss this in the original manuscript, we appreciate the reviewer bringing this up, and we are now addressing it. We will also add further details in the supplementary materials.
>
> **1. Boundary Conditions of the Radiation Field:**
> We assume Homogeneous Neumann Boundary Conditions for the radiation field, which implies that the gradient of the field at the boundary is zero:
> $\left. \frac{\partial L_s^i(x, y, \lambda)}{\partial n} \right|_{\partial \Omega} = 0$
>
> This assumption reflects the smooth transition at the boundary, which is commonly observed in natural and remote sensing images. It is a standard practice in image processing and remote sensing, as real-world scenes typically exhibit smooth transitions at boundaries [R8-R10].
>
> **2. Implicit Learning of Boundary Conditions and Explicit Spectral Responsivity Constraints.**
> Although we did not explicitly incorporate boundary conditions into the network architecture or loss function, the network could implicitly learn these boundary conditions.
> In supervised learning, networks can learn smooth transitions at the boundaries through the spatial smoothness enforced by the convolutional operations, especially with the use of padding operations.
> This behavior has been well-documented in the literature.
> For instance, networks trained on image restoration or remote sensing tasks have been shown to learn boundary smoothness even without explicit constraints [R11, R12].
>
> Furthermore, as illustrated in the Appendix, our visual analysis of the learned radiance fields demonstrates that the model indeed predicts smooth transitions at the image boundaries, which highly similar to the ground truth images.
> In addition, while the radiation field’s boundary conditions are implicitly learned by the network, we focus more on ensuring the **physical consistency of spectral responsivity**. Since spectral responsivity is **sensor-specific** and independent of the image content, we enforce **smoothness and non-negativity** constraints on the learned responsivity functions using a sigmoid activation function (Please refer to the Eq. 9 in the main paper).
>
> This function ensures that the spectral responsivity remains positive and smooth across the spectrum, aligning with physical sensor properties. The responsivity function is modeled through content-independent learnable parameters, providing the network with the flexibility to learn sensor-specific behaviors.
> Unlike the implicit boundary learning in the radiation field, the explicit constraint on the spectral responsivity is crucial because it governs sensor-specific characteristics that are independent of image content. This approach guarantees that the model’s learned spectral response adheres to realistic sensor constraints, ensuring the overall physical consistency of the method. As shown in Appendix A.4, our model learns radiance maps that preserve fine-grained structures and wavelength-dependent variations, as well as sensor responsivity curves that resemble real-world spectral responses, further supporting the physical consistency of our model.
>
> **Q3. Discretization sampling strategy in wavelength modulation:**
> In our current implementation, we adopt a *uniform sampling* strategy across the wavelength domain, as we do not make dataset-specific assumptions about spectral distribution and uniform sampling yields effective performance and results in our experiments.
> To enhance the clarity of our paper, we will emphasise this in our revised paper.
> The *adaptive sampling* strategies, for example, based on the second derivative of the sensor's spectral response curve to focus on regions with high spectral variability, could potentially enhance spectral representation. We consider this a valuable future direction.
>
> [R1] Zhou M, Huang J, Guo C L, et al. Fourmer: An efficient global modeling paradigm for image restoration[C]//International conference on machine learning. PMLR, 2023: 42589-42601.
>
> [R2] Hou J, Cao Z, Zheng N, et al. Linearly-evolved transformer for pan-sharpening[C]//Proceedings of the 32nd ACM international conference on multimedia. 2024: 1486-1494.
>
> [R3] Huang J, Chen H, Ren J, et al. A General Adaptive Dual-level Weighting Mechanism for Remote Sensing Pansharpening[C]//Proceedings of the Computer Vision and Pattern Recognition Conference. 2025: 7447-7456.
>
> [R4] He X, Cao K, Zhang J, et al. Pan-mamba: Effective pan-sharpening with state space model[J]. Information Fusion, 2025, 115: 102779.
>
> [R5] Deng S, Deng L J, Wu X, et al. Bidirectional dilation transformer for multispectral and hyperspectral image fusion[C]//Proceedings of the Thirty-Second International Joint Conference on Artificial Intelligence. 2023: 3633-3641.
>
> [R6] Deng S Q, Deng L J, Wu X, et al. PSRT: Pyramid shuffle-and-reshuffle transformer for multispectral and hyperspectral image fusion[J]. IEEE Transactions on Geoscience and Remote Sensing, 2023, 61: 1-15.
>
> [R7] Zhu C, Deng S, Zhou Y, et al. QIS-GAN: A lightweight adversarial network with quadtree implicit sampling for multispectral and hyperspectral image fusion[J]. IEEE Transactions on Geoscience and Remote Sensing, 2023, 61: 1-15.
>
> [R8] Dai S, Han M, Xu W, et al. Softcuts: a soft edge smoothness prior for color image super-resolution[J]. IEEE transactions on image processing, 2009, 18(5): 969-981.
>
> [R9] Li F, Wang Z, He G. AP shadow net: A remote sensing shadow removal network based on atmospheric transport and Poisson’s equation[J]. Entropy, 2022, 24(9): 1301.
>
> [R10] Weng C Y, Zhan Y, Gu X, et al. The Spatially Seamless Spatiotemporal Fusion Model Based on Generative Adversarial Networks[J]. IEEE Journal of Selected Topics in Applied Earth Observations and Remote Sensing, 2024, 17: 12760-12771.
>
> [R11] Liu S, Liu T, Gao L, et al. Convolutional neural network and guided filtering for SAR image denoising[J]. Remote Sensing, 2019, 11(6): 702.
>
> [R12] Zhu F, Liang Z, Jia X, et al. A benchmark for edge-preserving image smoothing[J]. IEEE Transactions on Image Processing, 2019, 28(7): 3556-3570.

---

> > ### Comment · Reviewer_3YGA · 2025-08-07
> >
> > Thanks for the author's rebuttal. The author has solved my questions in detail, so I will increase my score.

---

> ### Author Response · Authors · 2025-08-04
> **Follow-Up on Rebuttal Response**
>
> Dear Reviewer,
>
> Thank you once again for your valuable comments and the time you have devoted to reviewing our work. We have carefully addressed your concerns in detail and hope that you find our responses satisfactory, as other reviewers have. As the discussion phase is nearing its end, we would greatly appreciate any additional feedback you may have. We are always happy to communicate further and clarify any remaining concerns you might have.

---

> > ### Comment · Area_Chair_fLEN · 2025-08-07
> > **Please let us know what you think about the rebuttal.**
> >
> > Dear reviewer,
> >
> > You are the only negative reviewer for this paper. Did you have a chance to see the authors' rebuttal? Please let us know what you think and your opinion on this paper. Many thanks!
> >
> > Best,
> >
> > AC

---

### Note · Authors · 2025-08-13

We sincerely thank the reviewers for their insightful comments and constructive feedback on our submission, and we greatly appreciate the time and effort they dedicated to reviewing our work. These constructive comments and suggestions have significantly improved our manuscript, guiding us to further clarify the key concepts and implementation details. We are particularly encouraged by the positive feedback  highlighting the innovation of our method and its contributions to advancing physical interpretability in deep learning-based pansharpening techniques.

In response to the reviewers’ concerns and suggestions, we have diligently clarified the key aspects of our model, including the encoder architecture, the model’s physical interpretability, and the computational cost considerations. Our responses comprehensively address all raised concerns and enhance the methodological details. Importantly, our rebuttals have been met with positive recognition from each reviewer.

We also sincerely appreciate the assistance and guidance of the AC in facilitating this constructive dialogue during the rebuttal period, which ensures a fair and interactive review process, thereby greatly improving the quality of our work. We believe this work contributes new insights into the formulation of physically interpretable pansharpening frameworks by grounding model design in the sensor imaging process.

We look forward to the opportunity to contribute further to this field.

---

### Decision · Program_Chairs · 2025-09-17

**Decision:**

Accept (poster)

**Comment:**

The paper proposes a physics-informed neural operator framework for pansharpening that explicitly models the sensor imaging process, combining a spatial-spectral encoder, neural integral operator, and learnable spectral responsivity. Reviewers praised the novelty, strong empirical performance, and improved interpretability compared to prior methods. Concerns were raised regarding computational cost, theoretical depth of the physical modeling, and limited dataset scale, but these were addressed satisfactorily during rebuttal with clarifications on efficiency, boundary conditions, and extended analysis. Reviewers subsequently increased or confirmed their positive ratings. Overall, the contribution is technically solid, impactful for both remote sensing and physics-informed learning, and merits acceptance.